# Design and Modeling of Fiber-Free Optical MEMS Accelerometer Enabling 3D Measurements

**DOI:** 10.3390/mi13030343

**Published:** 2022-02-22

**Authors:** Samir Abozyd, Abdelrahman Toraya, Noha Gaber

**Affiliations:** Center for Nanotechnology, Zewail City of Science and Technology, October Gardens, Giza 12578, Egypt; snabozyd@zewailcity.edu.eg (S.A.); amtoraya@zewailcity.edu.eg (A.T.)

**Keywords:** optical accelerometer, MOEMS, microfabrication, 3D acceleration measurement

## Abstract

Optical accelerometers are popular in some applications because of their better immunity to electromagnetic interference, and they are often more sensitive than other accelerometer types. Optical fibers were employed in most previous generations, making micro-fabrication problematic. The optical accelerometers that are suitable for mass manufacture and previously mentioned in the literature have various problems and are only sensitive in one direction (1D). This study presents a novel optical accelerometer that provides 3D measurements while maintaining simple hybrid fabrication compatible with mass production. The operating concept is based on a power change method that allows for measurements without the need for complex digital signal processing (DSP). Springs hold the proof mass between a light-emitting diode and a quadrant photo-detector, allowing the proof mass to move along three axes. Depending on the magnitude and direction of the acceleration affecting the system, the proof mass moves by a certain amount in the corresponding axis, causing some quadrants of the quadrant detector to receive more light than other quadrants. This article covers the design, implementation, mechanical simulation, and optical modeling of the accelerometer. Several designs have been presented and compared. The best simulated mechanical sensitivity reaches 3.7 μm/G, while the calculated overall sensitivity and resolution of the chosen accelerometer is up to 156 μA/G and 56.2 μG, respectively.

## 1. Introduction

Almost all smart machines nowadays contain an accelerometer, including electronics, automotive, naval transportation, airplanes, robotics and prosthetic devices. It plays an essential role in navigation, stability, control and safety monitoring [1]. For instance, most smartphones have an accelerometer for navigation and detecting phone movement [2]. Most accelerometers consist of a proof mass suspended by springs. When acceleration is applied, the proof mass moves under the force caused by the acceleration, which activates a restoring force in the springs. A linear relation links the movement distance of proof mass to acceleration. Many transduction techniques, such as piezoresistive, piezoelectric, thermal, electrostatic and optical transduction, have been used to convert the displacement of the proof mass to a measurable signal, and hence measure acceleration [3]. The piezoresistive effect is the change in resistivity due to applied strain. Piezoresistive accelerometers have simple and cheap fabrication and an easy electronics circuitry interface. However, as resistivity also has a high dependency on temperature, piezoresistive accelerometers suffer from high thermal drift. This issue hindered their sensitivity and resolution [4]. The piezoelectric effect describes the phenomena of electrical charge production due to mechanical stress in some dielectric materials. The piezoelectric effect is linear and it is easy to measure the voltage difference caused by it. However, the deposition of piezoelectric material is not compatible with MEMS as most of them are bulky and can contaminate the MEMS fabrication facility. Moreover, the manufactured piezoelectric MEMS accelerometers have small sensitivity and can only achieve low resolution [5]. Thermal accelerometers are among the few types of accelerometers that have neither a proof mass nor springs. As they depend on the heat flow change with acceleration instead. This causes a temperature difference between two sensors that can be used to calculate acceleration. They are simple in fabrication and cheap. However, they have low resolution, low sensitivity and are highly sensitive to temperature changes [6].

Most accelerometers in the market are electrostatic because they are cheap, robust, easy to fabricate, simple to read, small in size and have moderate sensitivity [7]. Accelerometers, ideally, should have high sensitivity, high resolution and large bandwidth with the ability to filter the undesirable interfering frequencies. Some applications need ultra-accuracy. For instance, some measurements of environmental changes, such as seismic waves, have low amplitude and low frequency [8]. However, the sensitivity and resolution of electrostatic accelerometers are not enough for such applications due to the limited ability to measure capacitance. To illustrate, the noise coming from electrostatic interactions with the electronics interfacing circuit and surrounding environment decrease the performance of the device [9]. Furthermore, electrostatic accelerometers give faulty readings in applications that involve exposure to high magnetic fields, such as medical applications. Therefore, it is required to encapsulate electrostatic accelerometers with bulky and costly shielding materials [10,11]. Consequently, electrostatic accelerometers fail to get the required accuracy in applications requiring ultra-accurate measurements or susceptible to noise interference, such as meteorological verification. Additionally, using electrostatic accelerometers with magnetic shields in applications containing a magnetic source is expensive and consumes a large volume. The typical acceleration range for the aforementioned accelerometer types starts from 0.5 mG [12].

Optical techniques avoid some of the aforementioned disadvantages, especially the vulnerability to magnetic fields and electrostatic discharge (ESD). The electronics interfacing circuit will still have ESD susceptibility. However, the mechanical-optical transduction of the sensor itself will not be affected. On the other hand, accelerometers that depend on the electrical capacitance change or electrical resistance change are sensitive to ESD in both the sensing part and the electrical interfacing part. Numerous techniques based on optical methods have been presented in the literature, such as spectral interference, reflected light position, change in refractive index, evanescent waves and light intensity variation.

The spectral interference method was presented in a device based on fiber Bragg grating (FBG). The FBG element is embedded in the horizontal axis of a compliant material; the compliment material is fixed from one side and connected to a proof mass from the other side. When the device is subjected to acceleration, the inertial force of the mass will cause compression and expansion of the compliant material, which will vary the strain applied on the FBG element. As a result, the output wavelength of the FBG will change, which can be used to sense the acceleration. The resolution of this configuration was about 1 mG at 1 Hz movement [13]. Likewise, a Fabry–Perot interferometer was used as an accelerometer. When the proof mass moves under applied acceleration, the length of the Fabry–Perot cavity increases. The change in length causes a change in the light resonance wavelength in the Fabry–Perot interferometer. The resolution reached 8.5 µG for this device [14]. Another device based on spectral interference was demonstrated but used a nano-scale distributed dielectric Bragg mirror to create a one-dimensional photonic crystal. When the proof mass moves under applied acceleration, the dimensions of the photonic crystal change. Hence, a shift in the wavelength happens. This device attained a resolution of 1.037 µG [15]. A photonic-crystal nanocavity was utilized in another device. It is composed of two shaped photonic-crystal nanobeams, one connected to the bulk and the other to the proof mass. Upon displacement of the proof mass because of acceleration the nanocavity dimension changes, which makes a shift in the optical resonance frequency. This device achieved about 10 µg resolution at 1 Hz movement [16]. The mentioned spectral interference devices achieved high sensitivity and resolution. However, they are bulky and the alignment of fibers is crucial for them. Hence, the assembly process and the capability for hybrid fabrication employed in typical mass production processes may be troublesome. Moreover, they need DSP to interpret their output signal.

An additional technique based on tracking the position of reflected light also offers an interesting acceleration sensing method. Laser light is projected into the mirror deposited on the proof mass surface; and when it reflects, its position is screened on a grid of photodetectors. When the proof mass moves up and down under acceleration, the position of reflected light on the detectors’ grid changes. According to the authors, the device was able to perform in low frequency and showed high sensitivity, and showed a resolution of 16 G at 1 Hz movement. [17]. However, its sensitivity and resolution are size dependent. Therefore, the device is expected to be bulky. Moreover, it needs a sophisticated multiplexing circuit to read its signal. A different design based on reflection was also demonstrated. An LED light is coupled into a fiber and directed toward a V-shaped mirror, which splits light and directs it toward two photodetectors through two fibers. The V-shaped mirror is connected to a proof mass, when the proof mass moves under acceleration the differential light signal read by the two detectors will change and acceleration can be sensed. This device may achieve 78 mG resolution [18].

A different method used the change in refractive index in fiber core that is caused by fiber bending to sense acceleration. Researchers exploited the concept of fiber Bragg grating (FBG) and light interference to sense the change of position and acceleration of the proof mass. When acceleration is applied, the fiber bends. This causes a shift in reflected light position and increases the power lost in the cladding of the fiber, which can be sensed and converted to the corresponding acceleration value. The achieved resolution was up to 170 µG [19]. This design is expected to be vulnerable to fiber alignment and needs a DSP too. Other researchers applied the concept of controlling the wavelength and intensity by exploiting evanescent waves emerging from a waveguide. The device consists of a proof mass, a waveguide ring resonator surrounding the proof mass and a photodetector. When the proof mass moves under applied acceleration, the distance between the proof mass and the waveguide changes and disturbs the evanescent waves. Thus, the wavelength of light changes. This device needs a spectrometer to detect the shift of wavelength that happens due to acceleration. The typical resolving power of a high resolving benchtop spectrometer is about 0.05 nm. As the sensitivity of the accelerometer is only 0.0025 nm/G shift in wavelength, the estimated resolution of this setup is about 20 G [20]. This device is simple in fabrication and assembly. However, it has a very low resolution and a short acceleration range as evanescent waves vanish at a very tiny distance.

Light power variation provides a simple acceleration sensing technique. Abbaspour-Sani et al., 1995 [21] introduced a device based on the light intensity version, where they sandwiched a shutter between LED and photodetector. The light flux received by the photodetector from the LED changes with the lateral displacement of the shutter in terms of external acceleration. According to the author the device can operate with a resolution of less than 1 G. Likewise, Ahmadian et al., 2018 [22] proposed a design of an accelerometer with a graphene finger connected to a proof mass and positioned in the light path. When the proof mass moves, the graphene finger blocks more light. Due to the lack of enough optical information in the paper, and using only the mechanical information, the mechanical noise of the device is 1 mG. Hence, the resolution of this device is not expected to exceed this value. Similarly, Tang et al., 2019 [23] presented a device with almost the same optical concept, but they aligned the proof mass with the detector, opened a slit in the proof mass and used a quadrant photodetector (QPD) instead of an ordinary photodetector. When the proof mass moves under the acceleration effect, the intensity of light reaching the QPD through the proof mass slit will change. The resolution of this device was down to 81.6 nG at 0.1 Hz movement. Nevertheless, rather than using optical technologies, the enhancement of sensitivity was done by lowering the elastic coefficient, which necessarily leads to a reduction in bandwidth (below 0.1 Hz), dynamic range (below 1 mG) and other characteristics. While the mentioned light intensity variation techniques are relatively simple, cheap and small in size, they failed to get the high precision requirement, which is one of the main advantages of optical accelerometers [24,25].

The aforementioned optical accelerometers have one of the following drawbacks. Some need the coupling of the optical sources and detectors to optical fibers, which makes the device assembly difficult, hinders the ability for hybrid fabrication and makes them bulky. Some others need digital signal processing (DSP) or a complex electronic multiplexing circuit, or their sensitivity and resolution are size-dependent. While some do not offer the high sensitivity and resolution of optical accelerometers. Moreover, all of them are single axial (1D). Moving the proof mass by a 3D-sensitive spring system is a well-known area [26]; however, the optical transduction from the displacement to the electrical signal should be able also to distinguish the direction, which is not always feasible by some of the aforementioned techniques.

This paper proposes an optical tri-axial accelerometer based on the optical power change that shows a high resolution and sensitivity while maintaining easy fabrication, simple assembly and small size. The paper presents the design of the proposed accelerometer. It describes the design of the mechanical and optical parts. Additionally, it shows finite element analysis (FEA) COMSOL simulation for the main mechanical parameters, MATLAB modelling of optical parameters and total sensitivity.

## 2. Functional Principle and Design

The proposed accelerometer consists of three parts, a light source such as an LED, a MEMS part consisting of a mass-springs system and a QPD. The proof mass is suspended by four folded springs allowing it to move in 3D. Moving the proof mass up will decrease the light passing to all four detectors. Moving the proof mass down will make the light passing to all the detectors increase. Moving it right will make the left two detectors read more light power than the two right detectors and vice versa. The power variation technique can provide measurements that require minimal or no post-processing, with a simple fabrication and assembly. The sensor should also have high sensitivity and resolution. The below modelling and simulation provide the estimation to these values, and comparison with their counterparts in literature. Furthermore, it is expected that the size of the proposed device will be small compared to other optical accelerometers because it does not contain fibers to guide the light and sensitivity does not depend on size, as shown in Figure 1.

### 2.1. Mechanical Design and Modelling

To explain the design and modelling of the mechanical part of the accelerometer, a few introductory words are needed about the planned MEMS fabrication process. Fabrication of the mechanical part of the proposed accelerometers can be done on silicon on insulator (SOI) wafer, and only requires two masks, a structure layer mask, and a back etching mask. The silicon structure layer is patterned using optical lithography then etched using DRIE down to the oxide layer to create the springs-proof mass structure with straight walls. The wafer is then patterned from the other side and back etched down to the bottom side of the oxide layer. The exposed oxide layer is then etched using HF, or other etching suitable methods, to release the structure. This fabrication process is compatible with SOIMUMP^®^s MEMSCAP^®^’s process. While the proposed fabrication process is much simpler and cheaper, as it uses only two masks of the four masks available in SOIMUMP^®^s, it is an advantage that the device can be manufactured using a publicly available fabrication service with a shared cost between multi-users. To clarify, the compatibility with an available process will enable the fabless institutions to easily fabricate the device without the need to ask the fab companies for a more costly, customized fabrication process. Therefore, SOIMUMP^®^s design rules were employed. This process is conducted on a silicon on insulator SOI wafer with 400 µm handle thickness, 2 µm oxide thickness and 25 µm structure layer thickness. It is worth mentioning that such a process consists of four lithography-patterning steps, the two we have mentioned in the paragraph beginning, and two metal layer deposition and patterning steps [27]. The proposed fabrication does not need metal layers. Nevertheless, the two metal layers will not affect the MEMS structure, as the process permits omitting them just by not including their masks in the layout. After fabrication of the mechanical part, it can then be bonded with the LED die and the quadrant detector die using anodic bonding. Anodic bonding will provide an accurate, reliable and easy assembly of the three parts of the accelerometer.

Well-designed springs should carry the proof mass, provide high mechanical 3D sensitivity and low off-axis movement without suffering from plastic deformation. Figure 2a shows the proof mass springs system in the proposed optical accelerometer. Figure 2b–d shows the three different spring designs.

Mechanical sensitivity can be defined as the distance the proof mass moves in micrometers if an acceleration of 1 G is applied. Using the force equilibrium shown in Equation (1) where $F_{m}$ is force, k is the spring’s constant, x is displacement, *m* is mass and *G* is acceleration in Earth’s gravitational acceleration unit, the mechanical sensitivity $S_{m}$ of the accelerometer can be derived as in Equation (2) [28]:(1)$$F_{m}=kx=ma_{in\;G},$$
(2)$$S_{m}=\frac{x}{a_{in\;G}}=\frac{m}{k},$$

The resonance frequency $f_{r}$ of a mechanical system is defined as described in Equation (3). Additionally, the bandwidth (BW) of the accelerometer is practically taken below the frequency of resonance [18].
(3)$$f_{r}=\frac{1}{2\pi }\sqrt{\frac{k}{m}},$$

Increasing the sensitivity is preferable. However, it will be at the expense of decreasing the bandwidth. Therefore, sensitivity and BW are design specifications that depend on the application. Moreover, springs should be designed to have the intended spring constant value only in the direction of the applied acceleration, while having very high stiffness in the other directions. Hence, three-axis accelerometers are very tricky to mechanically design, as their springs should be sensitive to movement in the three axes. This multi-axis sensitivity leads to undesirable off-axis movement and rotational movement of the proof mass. These movements can cause errors in the accelerometer’s response. Therefore, springs should be designed to decrease these undesirable movements. Several designs of the springs are introduced in this work trying to find a trade-off between different advantages and disadvantages:

In design 1, shown in Figure 2b, the springs consist of folded beams that have natural immunity against both rotational and off-axis movement. Moreover, the beams have notches, which decrease the rotational movement as they resist the torque. Furthermore, the beams have different widths and appendages to increase the fabrication yield without affecting the sensitivity much. This design has very good resistance against the off-axis and rotational movement. However, it is found that it has a very small mechanical sensitivity. This small mechanical sensitivity will decrease the overall accelerometer sensitivity and will make it lose the main advantage of the optical accelerometer, which is the high sensitivity. Therefore, design 2, shown in Figure 2c, is introduced.

In design 2, the thickness of the beam is reduced to 2 µm, which is the minimum feature in the SOIMUMPs fabrication process. Furthermore, the notch shape in the middle is used only on the last folded pair in each of the four springs. This approach highly increases the mechanical sensitivity while maintaining the rotational and off-axis movement to acceptable values. However, from the analysis of the fabrication process, design 2 fabrication yield is expected to be low. This is due to the very thin and long springs. To clarify, mechanical stresses affect the structure in the steps of the fabrication process that include heating or releasing a layer. Therefore, thin, long beams cannot accommodate these stresses and tend to crack or break. This leads to the overall failure of the device. Hence, design 3 is introduced.

The springs’ design shown in Figure 2d is optimized to increase the fabrication yield and the immunity against rotational and off-axis movements, while maintaining the mechanical sensitivity at an acceptable value compared with design 1. This is achieved by making the beams thicker in some places and thin in others. This increases the springs’ immunity against the stresses caused by the fabrication process and protects the springs from breaking or cracking [27].

### 2.2. Optical Design and Modelling

Figure 3a shows an isometric drawing that explains the optical idea of the proposed accelerometer. Figure 3b is the top view of the optical concept of the accelerometer at its main six positions.

The first step of optical analyses is to define the light propagation regime concerning whether it is a near field or a far field. This can be determined from the Fraunhofer distance, which defines the limit distance of the near field and is given by Equation (4):(4)$$d_{near}<\frac{2D^{2}}{\lambda },$$
where D is the width or diameter of the emitting device, and λ is the peak wavelength [29]. In our case, D < 300 µm and λ = 850 nm, which results in d < 211 mm; and thereby, the proof mass and detector are placed in the near field. Thus, a near field analysis should be conducted. I. Moreno proposed an analytical method to model the behavior of square LED irradiance in the near field. Equations (5) and (6) are based on his work. The LED is modelled as a Lambertian source. An LED with high power and a 60 degree half angle is chosen. The SFH 4770S high-power LED should be used to get high sensitivity, while the wide half angle will allow decreasing the size of the device without affecting the vertical axis optical sensitivity as will be shown. The LED electrical current is tuned to make it operate at an optical power of 500 mW to avoid saturation of the photodetector. Equation (5) describes irradiance of the LED with angle:(5a)$$M=-\ln\left(2\right)/\ln\left(\cos\theta_{0.5}\right),$$
(5b)$$L_{0}=\frac{M+1}{2\pi M},$$
(5c)$$Ls=Lo\;\cos^{M}\theta_{s},$$
where $\theta_{0.5}$ is the half angle. Figure 4 shows the modelled radiant characteristics curve of the chosen LED, based on its high power and wide-angle, as will be discussed later, compared with the radiance plot in the LED datasheet to make sure the model can correctly describe the behavior of the chosen LED [30]. The maximum error between the two graphs is 17% at a 90° angle, while the average error is only about 4%.

Equation (6) describes the irradiance E in W/m^2^ of a square LED at a distance h in any point at xy [31]:(6)$$\begin{array}{c} E(x,y,h)=\frac{\Phi_{s}}{2\pi A_{s}}\{\frac{0.5*S_{y}-y}{\sqrt{{\left(0.5*S_{y}-y\right)}^{2}+h^{2}}}*\mathrm{atan}\left(\frac{0.5*S_{x}-x}{\sqrt{{\left(0.5*S_{y}-y\right)}^{2}+h^{2}}}\right)-\frac{0.5*S_{y}-y}{\sqrt{{\left(0.5*S_{y}-y\right)}^{2}+h^{2}}}*\\ \mathrm{atan}\left(\frac{-\left(0.5*S_{x}+x\right)}{\sqrt{{\left(0.5*S_{y}-y\right)}^{2}+h^{2}}}\right)+\frac{0.5*S_{x}+x}{\sqrt{{\left(0.5*S_{x}+x\right)}^{2}+h^{2}}}*\mathrm{atan}\left(\frac{0.5*S_{y}-y}{\sqrt{{\left(0.5*S_{x}+x\right)}^{2}+h^{2}}}\right)-\\ \frac{0.5*S_{x}+x}{\sqrt{{\left(0.5*S_{x}+x\right)}^{2}+h^{2}}}*\mathrm{atan}\left(\frac{-\left(0.5*S_{y}+y\right)}{\sqrt{{\left(0.5*S_{x}+x\right)}^{2}+h^{2}}}\right)\;\;-\frac{0.5*S_{y}+y}{\sqrt{{\left(0.5*S_{y}+y\right)}^{2}+h^{2}}}*\\ \mathrm{atan}\left(\frac{-\left(0.5*S_{x}+x\right)}{\sqrt{{\left(0.5*S_{y}+y\right)}^{2}+h^{2}}}\right)+\frac{0.5*S_{y}+y}{\sqrt{{\left(0.5*S_{y}+y\right)}^{2}+h^{2}}}*\mathrm{atan}\left(\frac{0.5*S_{x}-x}{\sqrt{{\left(0.5*S_{y}+y\right)}^{2}+h^{2}}}\right)+\\ \frac{-0.5*S_{x}+x}{\sqrt{{\left(0.5*S_{x}-x\right)}^{2}+h^{2}}}*\mathrm{atan}\left(\frac{-\left(0.5*S_{y}+y\right)}{\sqrt{{\left(0.5*S_{x}-x\right)}^{2}+h^{2}}}\right)-*\mathrm{atan}\left(\frac{0.5*S_{y}+y}{\sqrt{{\left(0.5*S_{x}-x\right)}^{2}+h^{2}}}\right)\}, \end{array}$$
where $\Phi_{s}$ is the flux in Watts, $S_{y}$ is the length of the LED, $S_{x}$ is the width and h is the distance away from the LED emitting surface. Figure 5 shows the LED irradiance at the surface with a 1 mm^2^ area away from the LED at different distances. The power inside a specific area at a distance h away from the LED surface can be calculated by integrating the irradiance over this area, as determined by Equation (7):(7)$$P_{area}=\iint_{x1}^{x2}Edxdy,$$

In our proposed device, the power received by the detector is less than the incident power by the amount shadowed by the proof mass. Therefore, this detected power is determined by Equation (8):(8)$$P_{Detected}=P_{total}-P_{proofMass},$$

The used detector is a quadrant detector. Thus, it is treated as four-isolated detectors. In addition, the proof mass is divided into four quarters to simplify the calculations and modeling.

One may question the choice of a high-power LED, which is not preferable, as this may cause heating and would consume much electrical power. Instead of that, a slit or more can be opened in the proof mass to allow more light to reach the detector and abandon the need for a high-power LED. Unfortunately, the typical distribution of the light intensity for any LED source is maximum and almost flat in the middle, and then gradually decreases on the border, as shown in Figure 5. The useful part for getting a change in the received light upon the mechanical motion is the gradient part, not the flat one. In addition, for this gradient part to still have an acceptable intensity to get high sensitivity, the high-power LED is essential. Hence, etching a big hole inside the proof mass or many small ones can make the device more power efficient by allowing more light to pass. Nevertheless, this power will have almost the same amount upon the proof mass shifting, as the intensity in the middle is flat, not gradient. Moreover, the fabrication of the device will be much more complex, as making a through-wafer hole inside the middle of the thick proof mass is non-reliable, requires a very high DRIE aspect ratio and can make the device brittle. Moreover, making holes inside the proof-mass will decrease the mechanical sensitivity and increase mechanical noise as the mass of the proof mass will decrease.

For selecting a suitable LED source to achieve good performance, there are some parameters to consider such as the irradiance and the spreading angle of the light. The narrow-angle LED is not good for the proposed accelerometer because it shows a very small slope (sensitivity) in the detected-power versus vertical distance curve for small vertical distance values. As shown in Figure 6a, which describes the behavior of a square LED with a 15° half-angle, the slope of the curve is flat until the distance between the LED and the proof mass becomes higher than 1 mm. This will require a big footprint of the device to get a good vertical sensitivity. Moreover, most narrow-angle LEDs in the market are low-power LEDs, which further decreases the sensitivity. On the other hand, wide-angle LEDs show a good vertical sensitivity at very small distances. Moreover, there are many wide-angle LEDs in the market with irradiance power higher than 1 W. For a square LED with 60° half-angle (SFH 4770S), the sensitivity is maximum and approximately linear when h<600 µm as, shown in Figure 6b. Thus, the proof mass should be placed at a distance smaller than 600 µm from the LED. Note that the LED is used in the system as a chip without packaging so that the h distance can be controlled. Then the entire system can be packaged together after chips’ assembly and bonding by a wafer bonder.

As for the QPD, the QP5.8-6 TO PIN is chosen because it has a stable responsivity with temperature, only 0.4 nA dark current at room temperature and 0.64 A/W peak responsivity. The noise-equivalent power (NEP) is only $1.8\times 10^{-14}\;W/\sqrt{\mathrm{Hz}}$, which enables high-frequency operation without decreasing the signal to noise ratio much. Additionally, the active area per quadrant is 1.44 mm^2^ which enables the photodetector to capture all light that bypasses the proof mass [32]. Moreover, the maximum quantum efficiency of the detector is aligned with the 850 nm peak wavelength of the chosen LED. To elaborate, the bandwidth close to maximum quantum efficiency usually has the most stable responsivity with temperature [33]. Furthermore, the 850 nm LED’s peak is very close to the 900 nm peak responsivity of the photo-detector, which will lead to high sensitivity.

## 3. Computational Results

### 3.1. COMSOL Simulation of the Mechanical Performance of the Mass-Spring System

Before discussing the simulation of the MEMS part, its dimensions should be mentioned. The proof mass and total chamber dimensions are 1 mm × 1 mm and 2 mm × 2 mm, respectively; the dimensions of the proof mass and springs are shown in Figure 2a. The thickness of the proof mass is 25 µm and the wafer thickness is about 400 µm.

To determine the mechanical performance expected from the proposed mass-spring systems, COMSOL is used to simulate the three different designs using solid mechanics physics. The stationary solver was used to calculate the proof displacement-acceleration relation. While the eigenfrequency solver was used to calculate the mechanical modes of the device. The 3D vertical and lateral displacements under 10 G acceleration for design 3 are shown in Figure 7. The proof mass displacement under the 10 G z-direction acceleration is about 14 µm. While the displacement of the proof mass under the 10 G x-direction acceleration is 18 µm. Acceleration was applied to each direction separately. To clarify, to calculate the displacement-acceleration relation in the z-direction, acceleration was applied in the z-direction only, and so on.

Figure 8 plots the relation between applied acceleration and the proof mass displacement in both vertical and lateral movement for the three designs. The mechanical sensitivities of the three spring designs are the slope of the graphs shown in Figure 8. Sensitivity for design 1 is small as expected and found to be, only, 0.25 µm/G for the vertical movement, which is given the symbol ($S_{Mv}$), and 0.05 µm/G for the lateral movement, which is given the symbol ($S_{Ml}$). While sensitivity for design 2 is 3.7 µm/G for the vertical movement, and 2.5 µm/G for the lateral movement, which is more than 50 times the sensitivity of design 1. As for design 3, the sensitivity is 1.4 µm/G for the vertical movement and 1.8 µm/G for the lateral movement, which represents a trade-off between the other two designs as explained earlier.

Figure 9 shows the modes of resonance for design 3. The spring constant is proportional to the mechanical resonance frequency according to Equation (3). Moreover, according to Equation (2), sensitivity is inversely proportional to the spring constant. Hence, it can be noticed that the lateral and vertical movements are more sensitive than the rotational and off-axis movements as they have smaller resonance frequencies. This means that the accelerometer has good resistance to rotational and off-axis movements.

### 3.2. Optical Sensitivity Modeling Results

To determine the optical performance, Equations (6)–(8) were implemented in MATLAB code. As explained earlier in the function principal section, if the acceleration on the proof mass is downwards to the negative z-direction, it will make the detected power increase in the four quadrants of the detector and vice versa, as shown in Figure 10. On the other hand, if acceleration is in the positive x-direction, it will make the back two detectors read more light power than the two front detectors and vice versa, as shown in Figure 11. The zero points in the figures are the stability points, which are corresponding to a 20 µm distance from the LED for the vertical direction and the middle point for lateral directions when the LED and the proof mass are aligned.

The vertical optical sensitivities for each quadrant of the detector are the slopes of the curves in Figure 10. Making a linear fitting for this curve, the resulting sensitivity for each one of the four detectors is $S_{Ov}$ = 0.233 mW, where
(9)$$P_{Detected}=0.233z+4.7\;\mathrm{mW},$$

The lateral optical sensitivities for each quadrant of the detector are the slopes of the curves shown in Figure 11. By making a linear fitting for this curve, the resulting sensitivity for each one of the four detectors, neglecting the slope sign, is $S_{Ol}$ = 0.123 mW, where
(10)$$P_{Detected}=0.123y+5.18\;\mathrm{mW},$$

### 3.3. Total Specification

#### 3.3.1. Total Sensitivity

To calculate the overall sensitivity of the device, the mechanical sensitivity should be multiplied by the optical sensitivity, then both multiplied by the mean photodetector responsivity over the LED FWHM spectrum (R), which is ~0.5 A/W. The vertical total sensitivity for each quadrant is:(11)$$S_{Tv}=S_{Mv}\times S_{Ov}\times R=0.156\;\mathrm{mA}/G$$

The lateral vertical total sensitivity for each quadrant is:(12)$$S_{Tl}=S_{Ml}\times S_{Ol}\times R=0.11\;\mathrm{mA}/G$$

#### 3.3.2. Current Range

A high acceleration range is not recommended for a high sensitivity accelerometer because there is a trade-off between sensitivity and acceleration range. To clarify, a high sensitivity accelerometer, like the proposed one, may suffer from non-linear behavior at high acceleration, as this high acceleration causes large displacement for the proof mass. This high displacement may lead to non-linear behavior in springs’ response, and in the optical sensitivity, as shown in Figure 6b. Hence, the acceleration range is chosen to be less than ±10 G. This corresponds to a displacement equal to $S_{Mv}\times acceleration\;range=\pm \;14$ μm. The corresponding optical power range that is detected by the photodetector can be calculated using Equation (9), which results in $P_{Detected}$ ranges from 7.96 to 1.438 mW. To calculate the output current range, the $P_{Detected}$ range is multiplied by the mean responsivity. Thus, it ranges from 0.72 mA to 4 mA.

#### 3.3.3. Noise and Resolution

Estimating the noise is a bit challenging as there are many sources of noise; each of them depends on many factors. In the proposed device, noise can originate from the mechanical part, the optical part or the electrical interface. Electronic interfacing circuit is outside the scope of this paper. For the mechanical part, the noise equivalent RMS-acceleration can be calculated from the following equation [18]:(13)$${noise}_{mechanical}=\sqrt{\frac{k_{B}\omega_{0}^{2}T}{m}}\times 1/G,$$
where $k_{B}$ is Boltzmann constant, T is temperature in Kelvin, $\omega_{0}=734\pi \;\mathrm{rad}/\sec$ is the resonance angular frequency and m=70 µg is the mass of the accelerometer. Working at room temperature, $T=300^{{}^{\circ}}\;K$, produces ${noise}_{mechanical}=56$ μG.

For the optical noise, it can be determined by the noise equivalent power (NEP) and shot noise [34] using the following two equations:(14a)$${\mathrm{noise}}_{\mathrm{NEP}}=NEP*\sqrt{f_{s}},$$
(14b)$${\mathrm{noise}}_{\mathrm{shot}}=\sqrt{2eIf_{s}},$$
where e is the electron charge, *I* is the current and $f_{s}$ is the sampling frequency or the frequency that is used to acquire data. Based on the Shannon–Nyquist theorem $f_{s}$ equals the double of the maximum frequency of operation. The maximum frequency in this case is determined by the maximum mechanical frequency. To assure a linear response, maximum mechanical frequency is chosen to be under two thirds of the resonance frequency. As presented in Section 3.1, the resonance frequency of the operating mode is 367 Hz. Thus, $f_{max}=\frac{2}{3}f_{r}=245\;\mathrm{Hz}$, and $f_{s}=490\;\mathrm{Hz}$. According to the detector’s datasheet, $NEP=1.8\times 10^{-14}\;W/\sqrt{\mathrm{Hz}}$. By substitution in Equation (14a), ${noise}_{NEP}$ is equal to $3.98\times 10^{-13}\;W$. To get the ${noise}_{NEP}$ in the unit of acceleration, we should multiply it by the responsivity of the photodetector, then divide it by the sensitivity of the accelerometer, which gives 0.74 nG. For shot noise, ${noise}_{shot}=0.8\;\mathrm{nA}$, which is equivalent to 5 µG accelartion. To get the total optical noise, ${noise}_{Optical}=\sqrt{{noise}_{shot}^{2}+{noise}_{NEP}^{2}}\approx 5$ μG. This value is lower than the mechanical noise by one more than order of magnitude, so the total noise is ${noise}_{total}=\sqrt{{noise}_{Optical}^{2}+{noise}_{mechanical}^{2}}=56.2$ µG. Therefore, the total noise is dominated by mechanical noise. The total noise can give a direct measure of the smallest measurable acceleration. Moreover, it also can be used to indicate the resolution, which is the minimum resolved difference between two acceleration measurements. The optical noise is almost negligible compared to the mechanical noise. If the electrical interfacing circuit causes a negligible amount of noise too, the calculated total noise value can be used as the accelerometer resolution.

As mechanical noise is the dominant noise type, this can give us room to decrease the optical power of the LED to reduce the power consumption without affecting the resolution much. Decreasing the optical power by one order of magnitude (to operate at 50 mW instead of 500 mW) increases the optical noise from about 5 μG to 16 μG. Thereby the total noise—and consequently the resolution—increases from 56.2 μG to only 58.2 μG. However, it will also decrease the sensitivity from 156 μA/G to about 15.6 μA/G. This may render the design of the electronic interfacing circuit a bit more complex, especially if the accelerometer is mainly intended for measuring low acceleration amplitudes. Nevertheless, this solution can provide a good compromise if the application requires decreasing the power consumption.

It is expected that there will be some additional errors due to the misalignment between the three parts of the accelerometer in the range of 1–2 µm. This may happen due to misalignment of chips’ assembly and bonding by the wafer bonder. Such expected misalignment values are very tolerable as the area of the LED is 1 mm × 1 mm, the proof mass is also 1 mm × 1 mm and each quadrant of the detector is 1.44 mm × 1.44 mm. Moreover, some errors may come from the scattering of light from the springs’ surface and device walls. However, these errors are not expected to hinder the device’s performance as they can be calibrated during device testing and subtracted from the signal.

## 4. Discussion

From the previous design and calculations, the resulted accelerometer specs for design 3 can be summarized as in Table 1.

For the final expected device size, two different sizes are estimated as mentioned in the table. The stated size 1 cm × 1 cm × 2 cm is to count for the simple biasing and readout circuits beside the packaging, or upon using off-the-shelf LEDs and QPD if they are not available in the die form. Upon the availability of the source and detectors on unpackaged substrates, the actual size of the sensing parts of the dies holding the LED, the mechanical micromachined structure and the detector after their anodic bonding is expected to be only about 1.5 mm × 4 mm × 4 mm.

Based on the specs shown in the table, the proposed accelerometer can achieve good performance in comparison with other types of accelerometers, regarding sensitivity, resolution and size. Therefore, it can occupy the area of application that needs better immunity to the magnetic field and is less susceptible to ESD or needs sensitivity and resolution higher than some other kinds of accelerometer, without suffering from the big size, and complex fabrication and assembly of the traditional optical accelerometer, especially for measuring in 3D. To elaborate, it is less bulky compared to other optical accelerometers, does not need optical fibers and enables hybrid fabrication and assembly on the wafer level upon the availability of LEDs and detectors in the dies form. Finally yet importantly, it has a simple output signal, which does not need DSP.

On the other hand, it consumes high electrical power because of the rather high-power LED. Furthermore, it still needs assembly and alignment like other optical accelerometers, but that can be done on the chip level, which is far less challenging than incorporating optical fibers. As every benefit comes with a drawback, the proposed optical accelerometers also have some challenges. For instance, the fluctuation in the electrical power source may cause fluctuation in irradiance of the high power LED, which may lead to error in acceleration reading. A power stabilizer circuit may be used to solve this challenge. Moreover, the high-power LED may cause some heating in the device, this heating may cause a thermal drift in the photo-detector responsivity and the responsivity curve may change [35]. Hence, acceleration reading may have some errors. This problem may be solved using an algorithm to compensate for the thermal drift effect.

## 5. Conclusions

This paper presented the design, mechanical simulation and optical modeling of an optical 3D accelerometer, based on power change with a relatively small size, simple hybrid fabrication and assembly compatible with mass production, and simple output signal that needs no DSP. The presented optical accelerometer can achieve sensitivity up to 0.156 mA/G and resolution of 56.2 µG with only 1.5 mm × 4 mm × 4 mm in size. Upon verifying the presented modeling and calculations experimentally, the proposed optical accelerometer is expected to exceed the performance of typical electrostatic accelerometers and some optical accelerometers too. Moreover, it has inherited better immunity against the magnetic fields and is less susceptible to ESD than electrostatic accelerometers. Furthermore, it is less bulky, simpler in fabrication and assembly than most traditional optical accelerometers and to our knowledge, it is the only intrinsic three-axial single-chip optical accelerometer. However, it is bulkier, higher in power consumption and has more complex fabrication and assembly than the electrostatic accelerometers. Therefore, the proposed optical accelerometer may fill the market gap for serving the applications that need better immunity to the magnetic field and ESD, and higher sensitivity and resolution than the electrostatic accelerometers and some other types as well, while maintaining the size and fabrication complexity to the minimum. Moreover, the proposed three-axial mass-spring system can be applicable to only some (not all) other accelerometers with optical readout, if their optical principle can distinguish between the different directions of motion.

## Figures and Tables

**Figure 1 micromachines-13-00343-f001:**
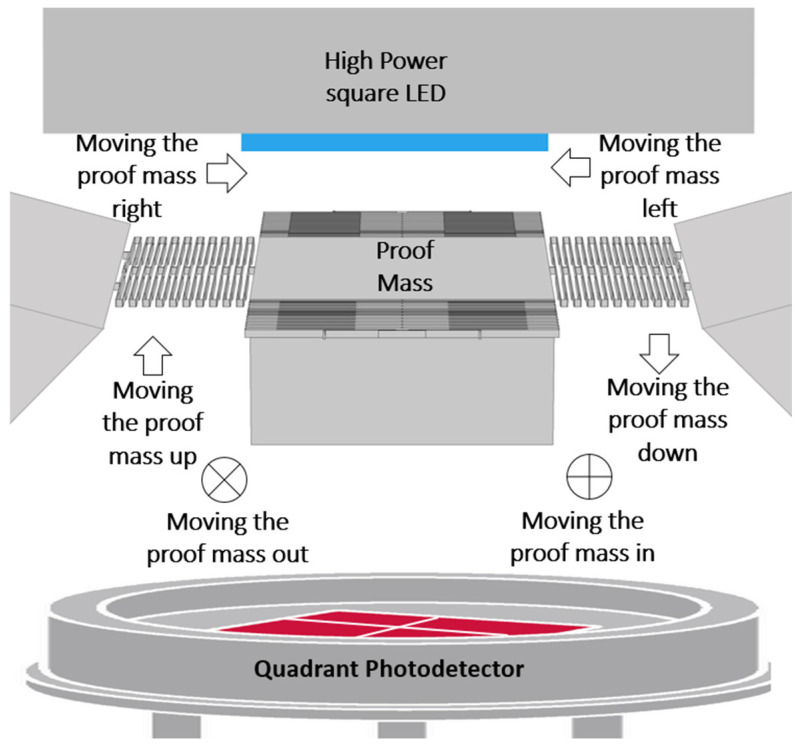
A schematic describes the proposed optical accelerometer working mechanism and parts. The dimensions are not to scale.

**Figure 2 micromachines-13-00343-f002:**
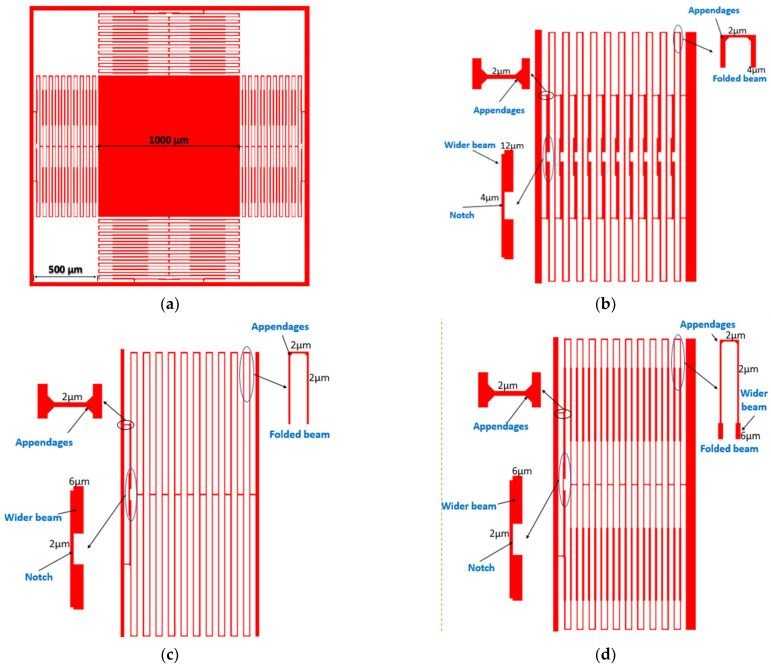
(**a**) Proof mass springs system. (**b**) Design 1: The most robust but least sensitive. (**c**) Design 2: The most sensitive but the most brittle. (**d**) Design 3: A compromise between sensitivity and robustness.

**Figure 3 micromachines-13-00343-f003:**
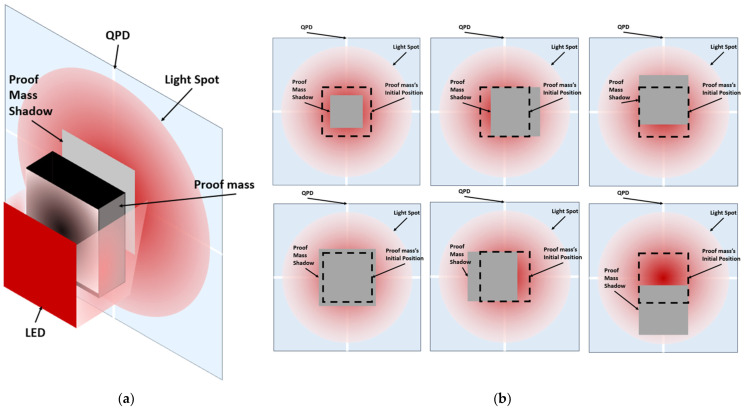
(**a**) Isometric drawing of the optical concept of the accelerometer. (**b**) The shadow position moves with the proof mass position and the optical power received by each quadrant of the QPD changes. The dimensions are not to scale.

**Figure 4 micromachines-13-00343-f004:**
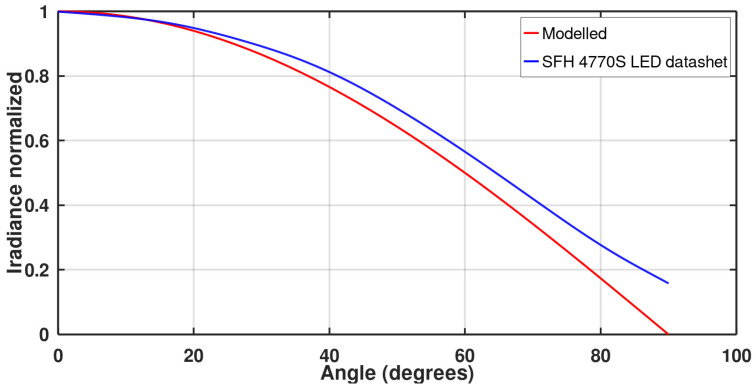
Comparison between the angle irradiance relation from the SFH 4770S LED’s datasheet and the modelled characteristics.

**Figure 5 micromachines-13-00343-f005:**
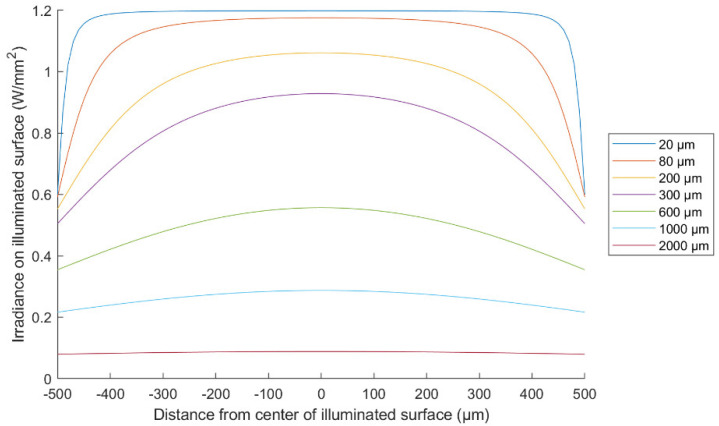
Square LED irradiance for several vertical distances projected on a square surface with a 1 mm^2^ area.

**Figure 6 micromachines-13-00343-f006:**
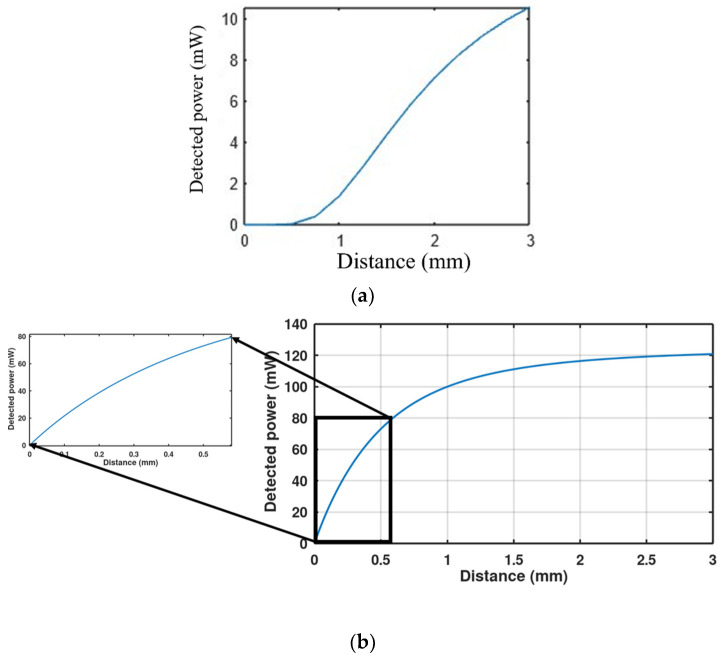
The change in power incident on the detector with the proof mass position distance from LED in the vertical direction for (**a**) a 15° narrow-angle LED, (**b**) a 60° wide-angle LED. The inset is a zoom on the short distance range.

**Figure 7 micromachines-13-00343-f007:**
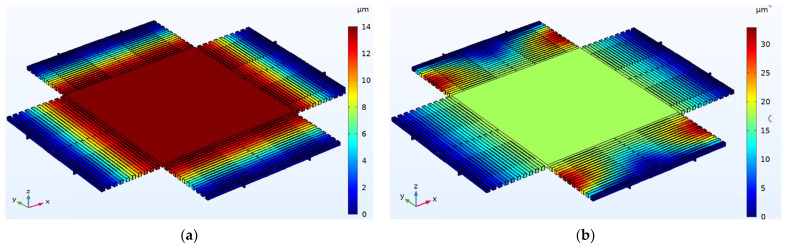
COMSOL simulation of design 3 for: (**a**) The vertical displacement (in z-direction) under 10 G acceleration, (**b**) the lateral displacement (in x-direction) under 10 G acceleration.

**Figure 8 micromachines-13-00343-f008:**
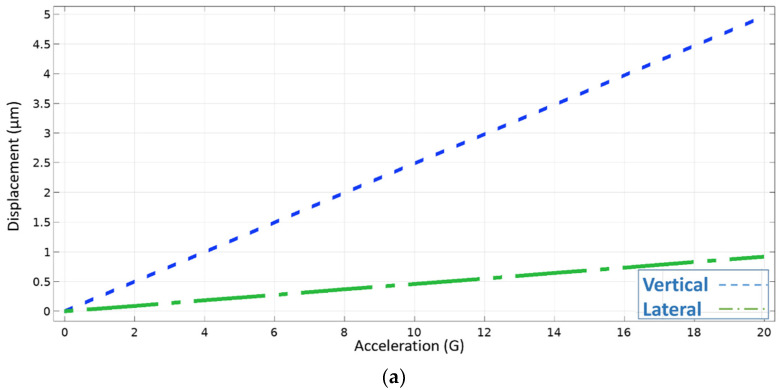
The relation between displacement and applied acceleration for each design is shown. The slope of each graph represents the vertical and lateral mechanical sensitivity of the three proposed optical accelerometer designs: (**a**) Design 1, (**b**) design 2, (**c**) design 3.

**Figure 9 micromachines-13-00343-f009:**
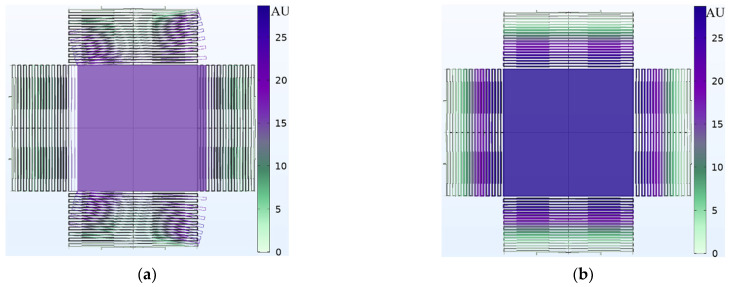
The mechanical resonance frequencies of the mass-spring system for design 3 (**a**) mode 1 lateral movement with resonance frequency 367 Hz, (**b**) mode 2 vertical movement with resonance frequency 428 Hz, (**c**) mode 3 rotational movement with resonance frequency 744 Hz, (**d**) mode 4 off-axis movement with resonance frequency 875 Hz.

**Figure 10 micromachines-13-00343-f010:**
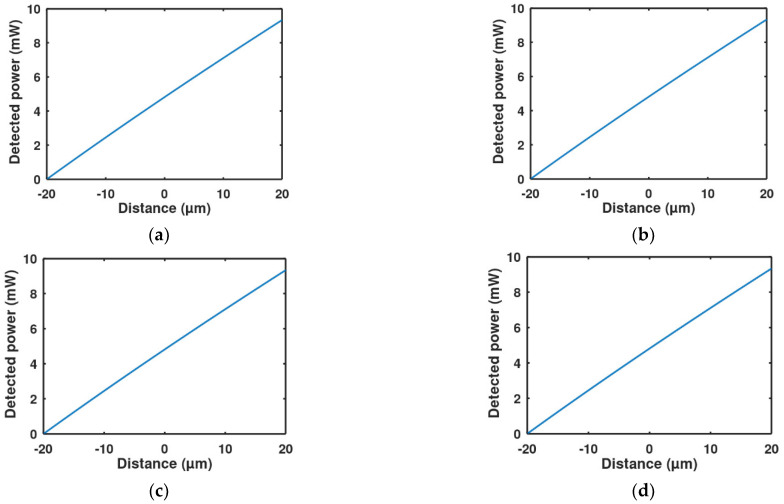
Detected power change with the vertical movement for each quadrant detector: (**a**) The left upper quadrant, (**b**) the right upper quadrant, (**c**) the left lower quadrant, (**d**) the right lower quadrant.

**Figure 11 micromachines-13-00343-f011:**
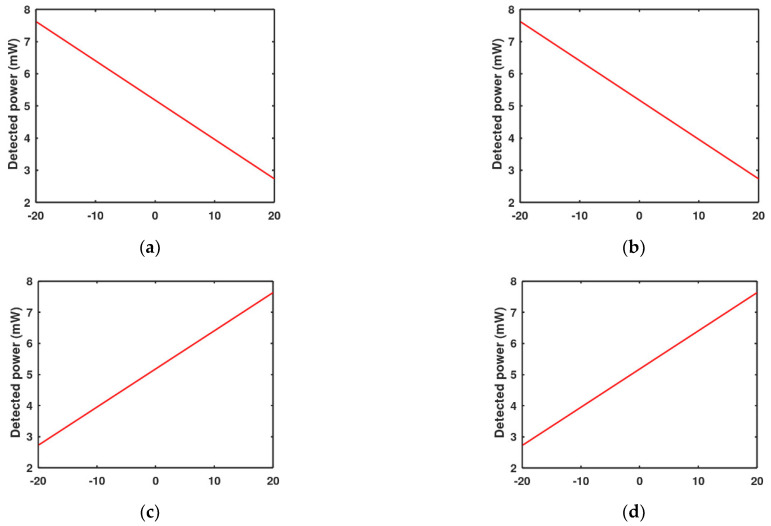
Detected power change with the lateral movement for each quadrant of the QPD: (**a**) The left upper quadrant, (**b**) the right upper quadrant, (**c**) the left lower quadrant, (**d**) the right lower quadrant.

**Table 1 micromachines-13-00343-t001:** Specs of the proposed optical accelerometer based on design 3.

Spec	Value
Current range	0.7 mA to 4 mA
Mechanical Bandwidth	244 Hz
Maximum acceleration	±10 G
Resolution	56.2 μG
Vertical Sensitivity	156 μA/G
lateral Sensitivity	110 μA/G
Orientation	axial, dual axial or tri axial
Size (anodic bonding)	1.5 mm × 4 mm × 4 mm
Size (off-shelf components)	1 cm × 1 cm × 2 cm

## Data Availability

The data presented in this study are available on request from the corresponding author. The data are not publicly available due to intellectual property rights.

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
