# Peer review of "Design and Modeling of Fiber-Free Optical MEMS Accelerometer Enabling 3D Measurements"

_micromachines, 2022, doi:10.3390/mi13030343_

Round 1

Reviewer 1 Report

The manuscript provides the description of the development and simulation of the accelerometer with optical measurement of the displacement of the proof mass. This is performed by evaluation of four signals from a quadrant photodetector.

The general idea of the work is easy to understand but many details remain unclear as it follows from the comments below. For this reason I can not consider this paper to be publishable in current form.

1. The title does not correctly represents the content of the paper. Looking at this title, an ordinary reader expects to see a presentation of the operational prototype of the proposed sensor, while in this manuscript I could find only a structural description of the suggested accelerometer, its theoretical analysis and computer simulation.

I also think that the term "real-time" in not needed in the title because the accelerometer is the device that by definition is supposed to provide real-time acceleration measurements.

2. In "Introduction" authors provide a brief review of various types of accelerometers. However, their comparison is quite obscure. The units that characterize the sensitivity of the accelerometers discussed in the introduction, as well as of that one proposed by the authors, should be unified in order to provide well defined comparison between devices based on different physical principles. Otherwise it is difficult to compare the characteristics of the sensors if the sensitivity is presented in non-comparable units: V/G, dB/mm, nm/G, mA/G, etc.

3. The work does not seem completed. In "Introduction", it was promised to present the fabrication steps of the accelerometer (see page 3, line 142), however, this information is missing.

4. In subsection 2.1, authors present a detailed description of spring mechanism and its behavior that may depend on the springs design. However, it is not clear what the materials are supposed to be used for springs and for the proof mass in the presented accelerometer. Is it plastic, metal or kind of composite material? It is quite strange that this important information is missing in the paper as it makes virtually impossible to imagine how this sensor can be fabricated and how reliable it might be in practical applications. This also makes difficult to understand, how its mechanical properties can be simulated in COMSOL without knowledge of the properties of materials.

5. In subsection 2.2, it is not clear where the equations (5) came from, and what is the meaning of the term "characteristic radiance" (see page 6, line 224). Also, it does not seem to be a good idea to use the same symbols for mass in equations (1-3) and an undeclared variable m in equations (5).

6. Further analysis of optical signal variations that depend on the position of the proof mass in the accelerometer is unclear. It might be helpful, for better understanding, to add a new figure showing the light propagating from the light source, a shadow on the photodetector surface formed by the proof mass and springs, and use this figure to explain the equations (5) and further graphics in figures 4 and 5.

7. What about light reflections from the springs and other parts of the accelerometer, and scattering from the internal walls of the chamber where the proof mass is located? What is their contribution in the measured optical signal? Do they introduce any errors in measured acceleration?

8. I have some doubts regarding the choice of the LED model as the light source. The optical power of this LED is about 2W but the expected detected power, as it follows from further analysis (see figures 9 and 10), is only about 50 mW per quadrant in the photodetector. Did authors try to find a better configuration to use the light more efficiently? In that case it might be possible to use a LED with lower power and avoid heating problems discussed in the "Conclusion" section.

9. The simulation of the propagation and detection of light by chosen photodetector model is unrealistic because the expected photodetector current range is 19-32 mA while the maximum current of QP5-6 TO is only 10 mA, therefore, in the operational conditions described in the paper, the detector will be saturated.

10. The evaluation of the photodetector noise is incorrect. Authors estimate this characteristic using the noise equivalent power (NEP) of the photodetector which is only valid for small input optical signal. At the power levels around 50 mW, the main source of the current fluctuations is a shot noise that is several orders of magnitude stronger than the noise defined from NEP. For example, if the photodetector current is about 30 mA (see page 14 of the manuscript), the shot noise within the bandwidth of 400 Hz can be estimated as 2 nA that is four orders higher than the noise calculated from NEP.

11. In the "Conclusions" section of the paper, authors claim that their proposed accelerometer provides higher sensitivity and resolution than the device based on the electrostatic effect. However, this claim can not be accepted simply because the proposed sensor is only in development phase, and even the experimental prototype still does not exist. While the authors provide a kind of theoretical analysis and some limited computer simulation, most of their concepts must be confirmed experimentally before making such a conclusion.

Recommended text corrections:

Page 2, line 84: What does mean "wavelength interference"?

Page 8, line 244: Replace "quadrature" by "quadrant".

Page 8, line 253: Replace "slop" by "slope".

Page 9, Figure 5b: The distance shown on the left plot has millimeter units and the exponent shows the order 10-4: does it really corresponds to 0.2 .. 0.6 microns?

Page 14, line 350: The mass unit nkg must be changed to μg.

Author Response

Response to the Reviewer’s Comments and Suggestions (Reviewer 1)

Dear Editors and Reviewers,

The authors would like to thank the reviewers for their knowledgeable criticism and wise recommendations towards the improvement of our manuscript. We are going to give answer to their comments and recommendation point-by-point and include the edits in the revised manuscript (Answers are typed in red).

My Best Regards,

Samir AboZyd

Comments and Suggestions for Authors (Reviewer 1)

  1. The title does not correctly represents the content of the paper. Looking at this title, an ordinary reader expects to see a presentation of the operational prototype of the proposed sensor, while in this manuscript I could find only a structural description of the suggested accelerometer, its theoretical analysis and computer simulation. I also think that the term "real-time" in not needed in the title because the accelerometer is the device that by definition is supposed to provide real-time acceleration measurements.

We meant by the term "real-time"  that the device does not require any DSP; the output is just electric currents that directly represent the acceleration. For many other optical accelerometers, for example, the Fabry-Perot and other light interference accelerometers, they require DSP circuitry and algorithms to analyze their output signal and extract the acceleration values from them. However, we removed this word from the title to prevent any misleading information. We also agree that the title should have stated that we only present the molding and design of the accelerometer. Therefore, we edited it in the revised manuscript to be “Design and Modeling of Fiber-Free Optical MEMS Accelerometer Enabling 3D Measurements".

  1. In "Introduction" authors provide a brief review of various types of accelerometers. However, their comparison is quite obscure. The units that characterize the sensitivity of the accelerometers discussed in the introduction, as well as of that one proposed by the authors, should be unified in order to provide well defined comparison between devices based on different physical principles. Otherwise it is difficult to compare the characteristics of the sensors if the sensitivity is presented in non-comparable units: V/G, dB/mm, nm/G, mA/G, etc.

We agree that unifying the physical units is essential for comparison. However, this is not an easy task for accelerometers sensitivity units, as every accelerometer has different physical characteristics and different output signal that cannot be represented by or converted to unified units. Thus, we provided the sensitivities in the units the other authors described in their papers. However, we can understand that some sensitivities are better than others by understanding the output signal. The output of the proposed accelerometer is current and we can, with a good ASIC circuit, measure current down to one nano Ampere. Therefore, with a sensitivity of 0.156 mA/G, the smallest signal that can be measured and the resolution of the proposed accelerometer will be hindered by noise only. To avoid any misunderstanding, we removed all the numerical values obtained by previous literature, so that the reader doesn’t get the notion that we are holding a quantitative comparison.

  1. The work does not seem completed. In "Introduction", it was promised to present the fabrication steps of the accelerometer (see page 3, line 142), however, this information is missing.

Indeed, we intended to put a section explaining the fabrication steps, but then we found that the SOIMUMS processes by MEMSCAP are suitable for such fabrication. So we intended to just refer to that in section 2, but apparently we missed that in our hurry to meet the deadline. We apologize for that mistake, and kindly find this info in the revised manuscript.   

  1. In subsection 2.1, authors present a detailed description of spring mechanism and its behavior that may depend on the springs design. However, it is not clear what the materials are supposed to be used for springs and for the proof mass in the presented accelerometer. Is it plastic, metal or kind of composite material? It is quite strange that this important information is missing in the paper as it makes virtually impossible to imagine how this sensor can be fabricated and how reliable it might be in practical applications. This also makes difficult to understand, how its mechanical properties can be simulated in COMSOL without knowledge of the properties of materials.

Kindly, pardon us for this forgetting. We use silicon and the details for materials used are described in the newly added parts in the subsection 2.1

  1. In subsection 2.2, it is not clear where the equations (5) came from, and what is the meaning of the term "characteristic radiance" (see page 6, line 224). Also, it does not seem to be a good idea to use the same symbols for mass in equations (1-3) and an undeclared variable m in equations (5).

We actually stated, the reference of both Equations 5 and 6 at the beginning " I. Moreno proposed an analytical method to model the behavior of square LED irradiance in the near field [30]."  For the characteristic radiance, we actually meant the characteristic of the irradiance, sorry for the mistake. We corrected it to be "irradiance" in the revised manuscript. 

The symbol (m) in equations (5) has been replaced by (M ) in the revised manuscript. We apologize for this mistake.

  1. Further analysis of optical signal variations that depend on the position of the proof mass in the accelerometer is unclear. It might be helpful, for better understanding, to add a new figure showing the light propagating from the light source, a shadow on the photodetector surface formed by the proof mass and springs, and use this figure to explain the equations (5) and further graphics in figures 4 and 5

The required figure is added as Fig. 3 in the revised manuscript.

  1. What about light reflections from the springs and other parts of the accelerometer, and scattering from the internal walls of the chamber where the proof mass is located? What is their contribution in the measured optical signal? Do they introduce any errors in measured acceleration?

It is expected that there will be some misalignment errors between the 3 parts of the accelerometer in the range of 2 µm.  This may happen due to misalignment of chips assembly and bonding by wafer bonder. Moreover, some errors may come from the reflection of light from the springs' surface. However, these errors are not expected to hinder the device's performance as they can be calibrated during device testing and subtracted from the signal. We could not combine the optical and mechanical parts of the device in one FEA simulation., as this needs a tremendous processing power and time. We actually tried multiphysics FEA using COMSOL, however, the simulation took days and then crashes without returning any outputs, which we repeated many times with different conditions and got the same errors.  This can be expected from a multiphysics FEA simulations as the number of points required for simulation is very massive. We need at least 4 points to represent one period of a sine wave. To simulate the propagation of light waves with 850 nm wavelength in about a 0.5 mm x 3 mm x 3mm volume, this will require 4.2 x 1016 points. Even using the less accurate ray optics will not make the processing power feasible. This is why we decided to separate the mechanical from optical modeling and used simplified analytical modeling for the optical part, as it is more complex from the processing point of view. We referred to such error in the revised manuscript and how it can be mitigated –along with other errors from misalignment during the wafers bonding- during the device calibration in the “3.3.3 Noise and resolution” subsection.  

  1. I have some doubts regarding the choice of the LED model as the light source. The optical power of this LED is about 2W but the expected detected power, as it follows from further analysis (see figures 9 and 10), is only about 50 mW per quadrant in the photodetector. Did authors try to find a better configuration to use the light more efficiently? In that case it might be possible to use a LED with lower power and avoid heating problems discussed in the "Conclusion" section.

We agree that the LED has a high power, which is not preferable as this may cause heating and would consume much electrical power. Additionally, based on your knowledgeable remark in this comment and comment 9, we decided to operate at 0.5W only. Unfortunately, the typical distribution of the light intensity from any LED source –including ours- is maximum and almost flat in the middle then gradually decreases on the border, as shown in figure 5 in the edited version. The useful part for getting a change in the received light upon the mechanical motion is the gradient part, not the flat one. And for this gradient part to still have acceptable amount of intensity to get high sensitivity, we had to select such high power LED. Indeed, we studied the idea of using other configurations and found that this one is perhaps the most suitable for our purpose of achieving a 3D, fabricable and high-performing optical accelerometer, although it may consume more power. For example, etching a big hole inside the proof mass or many small ones can make the device more power efficient by allowing more light to pass. But this power will have almost the same amount upon the proof mass shifting as the intensity in the middle is flat, not gradient. Also, the fabrication of the device will be much more complex, as making a through-wafer hole inside the middle of the thick proof mass is non-reliable, requires a very high DRIE aspect ratio, and can make the device brittle. Additionally, as far as we know, it is not compatible with the MEMSCAP process that we adopted. Moreover, making holes inside the proof-mass will decrease the mechanical sensitivity and increase mechanical noise as the mass of the proof mass will decrease.

  1. The simulation of the propagation and detection of light by chosen photodetector model is unrealistic because the expected photodetector current range is 19-32 mA while the maximum current of QP5-6 TO is only 10 mA, therefore, in the operational conditions described in the paper, the detector will be saturated.

We really thank you for this subtle recognition. To overcome this issue, we are going to operate the LED at 0.5 W, this can be done by operating at 0.5 A input current, as shown in the below figure from the LED’s datasheet. We will also place the proof mass closer to the LED at a 20 µm distance to compensate for the sensitivity reduction, and to decrease the amount of light projected onto the photo detector to avoid any saturation. Moreover, we should decrease the acceleration range from 20 G to 10 G because the proof mass will be closer to the LED and we wanted to avoid any touch of the proof mass with the LED with a 1.5x safe margin. After these edits the current range became from  to . Such corrections have been incorporated in the revised manuscript.  

  1. The evaluation of the photodetector noise is incorrect. Authors estimate this characteristic using the noise equivalent power (NEP) of the photodetector which is only valid for small input optical signal. At the power levels around 50 mW, the main source of the current fluctuations is a shot noise that is several orders of magnitude stronger than the noise defined from NEP. For example, if the photodetector current is about 30 mA (see page 14 of the manuscript), the shot noise within the bandwidth of 400 Hz can be estimated as 2 nA that is four orders higher than the noise calculated from NEP.

Thanks for the note, we edit this section and added the shot noise per your suggestion, and modified the total noise calculation accordingly, in the revised manuscript.

  1. In the "Conclusions" section of the paper, authors claim that their proposed accelerometer provides higher sensitivity and resolution than the device based on the electrostatic effect. However, this claim can not be accepted simply because the proposed sensor is only in development phase, and even the experimental prototype still does not exist. While the authors provide a kind of theoretical analysis and some limited computer simulation, most of their concepts must be confirmed experimentally before making such a conclusion

Thanks for drawing our attention to this. We edited the conclusion in the revised manuscript as per your advice.

  1. Recommended text correction

Page 2, line 84: What does mean "wavelength interference"?

  • For the wavelength interference, we mean the spectral interference, which happens in optical interferometers such as Fabry-Perot, or in photonic crystal structures, as detailed after it. We changed it to spectral interference.

Page 8, line 244: Replace "quadrature" by "quadrant".

  • "quadrature" is replaced by "quadrant"

Page 8, line 253: Replace "slop" by "slope".

  • "slop" is replaced by "slope"

Page 9, Figure 5b: The distance shown on the left plot has millimeter units and the exponent shows the order 10-4: does it really corresponds to 0.2 .. 0.6 microns?

  • There was a mistake in the Figure 5b and we changed it especially with the new optical power calculations

Page 14, line 350: The mass unit nkg must be changed to μg.

  • mass unit nkg is changed to μg

Reviewer 2 Report

The manuscript is dedicated to theoretical consideration and numerical simulation (by Comsol Multiphysics and Matlab) of several variations of a conventional MEMS accelerometer with a seismic mass held by rather complex micromachined springs, illuminated by an infrared LED diode positioned above the seismic mass and with a quadrant PIN photodiode mounted below the mass to detect the seismic mass movements. Such an overly simplified layout leads to quite a large device, 1 cm × 1 cm × 2 cm encapsulation, requires mechanical mounting and alignment of its parts instead of monolithic integration. The most original part is the approach to the design of the support springs. Scientifically, the whole design and simulation procedure is mostly sound, albeit there is a number of unsupported and even directly erroneous claims.

Besides being rather complex for practical implementation, albeit conceptually simple, the described main approach is actually almost 30 years old and obsolete among numerous superior solutions. Thus the question of the original contribution and novelty is posed. Any verifications of the obtained theoretical results are completely missing. There is a number of erroneous and misleading claims. Thus in my opinion the paper is unsuitable for publication – a decision that was hard for me to bring, since it is visible that a lot of effort has been invested in rendering the article. I think that even a thorough and detailed major revision could not help, since I fail to see how the originality could be improved and how the contribution can be moved beyond an incremental and very slight one. A detailed elucidation of my assessment follows.

1. Even though the sensing parts are often indispensable element of micromachines, the accelerometers in the present article have comparably huge overall dimensions (to quote the authors: “Size: 1 cm × 1 cm × 2 cm” – Table 1) which is hardly convenient for their use in any micromachines. The state of the art sensors after encapsulation typically measure 1 mm × 1 mm × 2 mm or less and cost below 0.5 € per piece. MDPI Sensors journal would be a more convenient venue for such kind of article and the mentioned sensor size if the article itself were qualitatively correct and presented sufficiently novel results.

2. Regretfully, most of the overall article quality criteria are not satisfied, including the most basic one – originality. The proposed photonic accelerometer design is conceptually trivial and has been known for quite a long time. Similar designs have been in use for almost three decades. More particularly, a concept of a proof (or seismic, or test) mass on supporting beams or springs, illuminated by a separate LED diode and with the modulated light signal detected by separately mounted photodetector had been published in 1995 (Abbaspour-Sani et al., “A novel optical accelerometer,” IEEE Electron Device Lett 16, 166–168, 1995.) If one has a very simple idea, one has to carefully double-check if somebody already proposed a similar device because there is a large chance for that, especially in such a vast, well-developed, sophisticated and ubiquitous field. For alternative optical approaches in older references, the authors could check e.g. the 1996 article by Berkoff and Kersey, “Experimental demonstration of a fiber Bragg grating accelerometer,” IEEE Photon. Technol. Lett. 8, 1677–1679, 1996; also Noell, W. et al. “Applications of SOI-based optical MEMS”. IEEE J. Sel. Top. Quantum Electron. 8, 148–154, 2002 or even Krause et al, “A high-resolution microchip optomechanical accelerometer”, Nature Photonics, 2012.

3. Obsoleteness: Related to the obsoleteness of the proposed design, the authors are advised to search literature more thoroughly and, if nothing, look for some of the existing reviews featuring optical/photonic accelerometers (e.g. Lu et al. “Review of micromachined optical accelerometers: from m g to sub-μ g”. Opto-Electron Adv 4, 200045 (2021) . doi: 10.29026/oea.2021.200045 or Wang et al., “Micromachined Accelerometers with Sub-µg/√Hz Noise Floor: A Review,” MDPI Sensors 2020, 20, 4054 or B. Malayappan, P. K. Pattnaik, “Optical MEMS Accelerometers,” chapter in book Microelectronics and Signal Processing, ed. Sanket Goel, CRC Press, 2021.) A number of articles on general MEMS accelerometers have already been published in MDPI Micromachines, including a special issue with 16 papers.

4. Flaws related to the actual fabrication: The presently proposed approach suffers from a number of practical flaws which, as far as I have seen, have not been dealt with by its authors, including the problems with accurate alignment of the three main physically separate but interdependent blocks (which is difficult to batch process, making the assembly a serious production bottleneck). Earlier in their text, the authors themselves eliminated some types of accelerometers because (to quote their own words “this design suffered from difficult assembly and complex fabrication.”) In addition to the alignment problems, quadrant photodiodes can be quite expensive and require a high accuracy, so this introduces a problem with the cost of the whole setup. The question is posed what is the point of going to such lengths if one could use for instance a simple Bragg structure instead and end with a higher accuracy and improved simplicity at a fraction of the cost – or even the much simpler and even less expensive capacitive or piezoresistive device which additionally offer the virtue of monolithic integration, in spite of their sensitivity to EM disturbances. Finally, there is a question how feasible is to fabricate the complex structure of the spring supports with acceptable quality, uniformity and yield.

5. Verification of the results is completely missing. The authors did not perform even a simplest proof of concept experiment to see if their concept is feasible and if the results are at least similar to the simulated ones, let alone a full experimental characterization of a finished and encapsulated device. They did not present a comparison with the literature data on similar structures (which do exist, as we have witnessed above).

There is a number of scientifically dubious claims and inconsistencies. I quote here only a few:

6. Incorrect claims: the authors express their opinion that fiber-based solutions are inferior to their approach since they are too large (albeit their own solution is actually cm-sized) and since fibers will bend under acceleration (they do not mention that fibers can be immobilized – e.g. by gluing to a substrate and that, alternatively, waveguides can be monolithically integrated into the same chip as the rest of the device.

7. Incorrect claims: The authors write that there are few optical applications of accelerometers, while in reality there is a vast number of them, especially among those presented within last few years (the authors are advised to check the above mentioned reviews from 2020 and 2021)

8. Incorrect claims: the authors write “it is expected that the size of the proposed device will be small compared to other optical accelerometers.” However, according to the authors themselves, the device dimensions are 1 cm × 1 cm × 2 cm. As far as the optical/photonic accelerometers are concerned, a majority of the photonic crystal-based devices are vastly smaller, while the standard, of-the-shelf non-optical devices usually take up a package of about 1 cubic millimeter.

9. Incorrect claims: the value of mechanical sensitivity (in mm/G or nm/G) is not the main figure of merit for a photonic accelerometer as the authors appear to think when they claim that the lower values of this parameter represent a disadvantage. Actually the overall sensitivity and device output will strongly depend on the readout method (for instance, interferometric readouts will effortlessly measure sub-nanometer displacements, so they will be able to utilize orders of magnitude smaller mechanical sensitivities and still obtain a high overall response).

10. Incorrect claims: the authors write that their accelerometer “is monolithic in fabrication and assembly,” while actually it is a false information, since commercial components are used for the light source and for the detector, which definitely are not monolithically integrated with the rest of the accelerometer (seismic mass).

11. Inconsistencies: At the beginning of their article the authors argue that the competing piezoresistive accelerometers “suffer from thermal drift” and quote this as their main deficiency (lines 44, 45). However, for their own sensor they quote, “high-power LED may cause some heating in the device, this heating may cause a thermal drift in the photo-detector responsivity and the responsivity curve may change [25]. Hence, acceleration reading may have some errors.” (lines 403-405) – basically the same issue they quoted to eliminate vastly cheaper and simpler piezoresistive and capacitive devices.

12. Inconsistencies: Why use Comsol Multiphysics to simulate mechanical properties only, while the optical properties are determined by Matlab, when the both simulations can be done simultaneously in Comsol (after all, this is the main advantage of this FEM package – even its full name includes the word “Multiphysics”).

13. Dubious claims: it appears that the authors implicitly assume that there are no scattering losses and no multiple reflections (e.g. from the walls, top and bottom of the encapsulating structure), while actually this could well be the main source of inaccuracies in their simulation. If they performed a full FEM simulation for the chosen geometry without any simplifications/approximations, they would have better data. I can only assume that such a simulation would necessarily be overly CPU-intensive.

14. Article organization: the manuscript is overlong and needlessly repeats the well-known facts. The long text about the non-optical solutions could be almost completely skipped (especially in the vastly overlong Introduction, which, paradoxically, quite weakly handles the optical/photonic devices which are the main topic of the paper. Introduction reads like a very short review paper on conventional MEMS accelerometers. There is no need for the description of the operating principles of every single type of non-optical MEMS accelerometers since none of it contributes to further reading. Only the essentials and pro et contra arguments for their use should be kept. A glaring example of needless inclusion of excess material is met already in Abstract, where the authors elucidate in no less than 3 lines the basic principles of the operation of quadrant PIN photodiodes.

15. English is rather readable throughout. This does not mean that it is without its share of errors in terminology and style. The authors should use “quadrants” instead of “panels” or “quarter detectors” for their quadrant photodiode. Also, they should not use the term “quadrature detector” (in a rather poorly written sentence “The used detector is a quadrature detector. Thus, it is treated as a four-isolated detector.”). They should rewrite “small slop”, “furtherly,” “ain,” “are be,” “a four-isolated detector,” “letral,” “mechnicle,” (repeated twice) “negligted,” “neglable,” “algorism,” etc.

16. The source of eq. 6 is not cited. Any descriptions, derivations or clarifications of eq. 6 are missing.

17. Nothing is mentioned about the existing solutions for 3D-sensitive spring systems in the state of the art part of Introduction, albeit this is a well thread and well-known area.

18. There is an error in Fig. 5: the two images are overlapped with what I expect was intended to be the inset.

19. References: the article does include recent publications, however these are not the most relevant ones. Also, disproportionally many of them are dedicated to the conventional and well known MEMS accelerometers. For more recent references on photonic accelerometers, authors are advised to see the above quoted review papers, and the references cited therein.

In conclusion, albeit the paper is painstakingly written and well analyzed, it is undermined by the unoriginal main assumption and by a lot of faulty judgments and inconsistent claims. Maybe it would have helped if at least some of the results were verified by experimental data.

Author Response

Response to the Reviewer’s Comments and Suggestions (Reviewer 2)

Dear Editors and Reviewers,

The authors would like to thank the reviewers for their knowledgeable criticism and wise recommendations towards the improvement of our manuscript. We are going to give answer to their comments and recommendation point-by-point and include the edits in the revised manuscript (Answers are typed in red).

My Best Regards,

Samir AboZyd

Comments and Suggestions for Authors (Reviewer 2)

  1. Even though the sensing parts are often indispensable element of micromachines, the accelerometers in the present article have comparably huge overall dimensions (to quote the authors: “Size: 1 cm × 1 cm × 2 cm” – Table 1) which is hardly convenient for their use in any micromachines. The state of the art sensors after encapsulation typically measure 1 mm × 1 mm × 2 mm or less and cost below 0.5 € per piece. MDPI Sensors journal would be a more convenient venue for such kind of article and the mentioned sensor size if the article itself were qualitatively correct and presented sufficiently novel results.

Thank you for drawing our attention to this point as we discovered that it might need further clarification. Actually, the stated size 1 cm × 1 cm × 2 cm was an exaggeration to count for the simple biasing and readout circuits beside the packaging, or upon using off-the-shelf LEDs and QPD if they are not available in the die form. Upon the availability of the source and detectors on unpackaged substrates, the actual size of the sensing parts of the 3 dies holding the LED, the mechanical micromachined structure, and the detector after their anodic bonding is expected to be about 1.5mmx4mmx4mm. We have clarified this point in the revised manuscript.

Indeed, there are accelerometers in less size, but they are of other types, not optical. Usually, they are based on electrical transduction techniques, such as electrostatic, piezoresistive, or piezoelectric; which are more suitable to ESD and electromagnetic interference than the optical ones. The other optical types –as discussed in the intro and in section 4 in the manuscript- usually incorporate optical fibers and are only 1D, while ours in 3D. So, the comparison should be between ours and 3 of the other optical types. Not mentioning the need of coupling the optical sources and detectors to the fiber, while ours has the source and detector already integrated. The image below present one of those optical type in the market; and as can be noticed, the optical fibers, and their 3D configuration is not as a single microchip.  

Figure 1: Luna’s os7500 accelerometer series [https://lunainc.com/sites/default/files/assets/files/resource-library/Distributed%20Vibration%20Monitoring%20using%20FO%20Accelerometers.pdf. Accessed: 22 Jan. 2022].

  1. Regretfully, most of the overall article quality criteria are not satisfied, including the most basic one – originality. The proposed photonic accelerometer design is conceptually trivial and has been known for quite a long time. Similar designs have been in use for almost three decades. More particularly, a concept of a proof (or seismic, or test) mass on supporting beams or springs, illuminated by a separate LED diode and with the modulated light signal detected by separately mounted photodetector had been published in 1995 (Abbaspour-Sani et al., “A novel optical accelerometer,” IEEE Electron Device Lett 16, 166–168, 1995.) If one has a very simple idea, one has to carefully double-check if somebody already proposed a similar device because there is a large chance for that, especially in such a vast, well-developed, sophisticated and ubiquitous field. For alternative optical approaches in older references, the authors could check e.g. the 1996 article by Berkoff and Kersey, “Experimental demonstration of a fiber Bragg grating accelerometer,” IEEE Photon. Technol. Lett. 8, 1677–1679, 1996; also Noell, W. et al. “Applications of SOI-based optical MEMS”. IEEE J. Sel. Top. Quantum Electron. 8, 148–154, 2002 or even Krause et al, “A high-resolution microchip optomechanical accelerometer”, Nature Photonics, 2012.

Actually, we made a thorough literature review for filing a patent for this design before writing this paper. As it is well known, a patent is far more demanding than a scientific publication regarding the originality of an idea. We carefully checked that no one ever has combined the advantages of being 3D, no complex DSP, integrability into a single chip or module by monolithic fabrication. We just didn’t put all the publications that we have encountered because this is not a review paper, so we just stated some examples to not be overly long. Indeed, there are accelerometers in literature based on the same concept of intensity variation, and we already cited some of those in the manuscript such as ref. 22 in the revised manuscript. The basic physical phenomena on which accelerometers can rely on are numbered –as stated in the introduction- and one can find several designs under each of them. Simplicity, or some may call trivial, is not something bad that should be avoided if it can do the job. On the contrary, the good engineer should find the solution by the easiest way. Especially to come up with a design that combine as much advantages as possible, beside overcoming the manufacturing challenges that probably hindered such solutions from reaching the market. The previous designs in literature –including those you kindly mentioned- were not 3D, and some of them needed a lens, a fiber or other components that hinders their direct assembly inside the cleanroom by its standard tools, while ours can be -simply- bonded by a wafer bonding (upon the availability of free standing dies for the source and detector). In addition, some of these alternative optical approaches also require sophisticated processing of the measured data to obtain the acceleration values, and even requires a non-integrable lab setup, such as those based on measuring the spectrum. So such approaches can not be developed into a practical solution in the form of a stand-alone module that can be directly integrated within the needed application requiring just electrical connections; while our approach can offer that.

The mentioned references have been cited in the revised manuscript.

  1. Obsoleteness: Related to the obsoleteness of the proposed design, the authors are advised to search literature more thoroughly and, if nothing, look for some of the existing reviews featuring optical/photonic accelerometers (e.g. Lu et al. “Review of micromachined optical accelerometers: from m g to sub-μ g”. Opto-Electron Adv 4, 200045 (2021) . doi: 10.29026/oea.2021.200045 or Wang et al., “Micromachined Accelerometers with Sub-µg/√Hz Noise Floor: A Review,” MDPI Sensors202020, 4054 or B. Malayappan, P. K. Pattnaik, “Optical MEMS Accelerometers,” chapter in book Microelectronics and Signal Processing, ed. Sanket Goel, CRC Press, 2021.) A number of articles on general MEMS accelerometers have already been published in MDPI Micromachines, including a special issue with 16 papers.

In our humble opinion, one can hardly call something obsolete if, until recently, researchers in the field are still interested in such designs. A close 1D design based on the same concept has been purplish on 2019 [Tang, S.; Liu, H.; Yan, S.; Xu, X.;Wu,W.; Tu, L.C. A MEMS Gravimeter Qualified for Earth Tides Measurement. In Proceedings of the 20th International Conference on Solid-State Sensors, Actuators and Microsystems (TRANSDUCERS), Berlin, Germany, 23–27 June 2019; pp. 499–502. - Tang, S.; Liu, H.; Yan, S.; Xu, X.;Wu,W.; Fan, J.; Liu, J.; Hu, C.; Tu, L. A high-sensitivity MEMS gravimeter with a large dynamic range. Microsyst. Nanoeng. 2019, 5, 1–13.]. Our design even surpasses them by being 3D, so our design can be equivalent to 3 of them.

The mentioned references have been cited in the revised manuscript.

  1. Flaws related to the actual fabrication: The presently proposed approach suffers from a number of practical flaws which, as far as I have seen, have not been dealt with by its authors, including the problems with accurate alignment of the three main physically separate but interdependent blocks (which is difficult to batch process, making the assembly a serious production bottleneck). Earlier in their text, the authors themselves eliminated some types of accelerometers because (to quote their own words “this design suffered from difficult assembly and complex fabrication.”) In addition to the alignment problems, quadrant photodiodes can be quite expensive and require a high accuracy, so this introduces a problem with the cost of the whole setup. The question is posed what is the point of going to such lengths if one could use for instance a simple Bragg structure instead and end with a higher accuracy and improved simplicity at a fraction of the cost – or even the much simpler and even less expensive capacitive or piezoresistive device which additionally offer the virtue of monolithic integration, in spite of their sensitivity to EM disturbances. Finally, there is a question how feasible is to fabricate the complex structure of the spring supports with acceptable quality, uniformity and yield.

Actually, the easy assembly of this design and its compatibility with the microfabrication was the main reason behind its choice. The 3 blocks can be fabricated on 3 different wafers, and then anodically bonded together by a typical wafer bonder, which is a common technique in MEMS fabrication.  Many bonders can offer high alignment accuracy, such as BA8 Gen4 that offers up to 0.25 μm for top-side alignment and 1 μm for bottom-side alignment [SUSS MicroTec BA8 Gen4 Pro Bond Aligner Available online: https://www.suss.com/en/products-solutions/wafer-bonder/ba8-gen4-pro (accessed on 20 January 2022).]. Even less sophisticated systems such as EVG®610 BA is ± 2 µm for backside alignment, and ± 1 µm for transparent alignment [EVG®610 BA Semi-automated Bond Alignment System Available online: https://www.evgroup.com/products/bonding/bond-alignment-systems/evg610-ba/(accessed on 20 January 2022)]. This is very convenient for our case as the area of the LED is 1 mm x1 mm, the proof mass is also 1 mm x1 mm, and each quadrant of the detector is 1.44 mm x1.44 mm. Even the small errors arises from such misalignment can be mitigated during the device calibration.

On the other hand, how to implement a single chip or module for a measurement setup based on a Bragg structure that implies phase shift measurements? How to integrate the entire lab-scale components used to transduce the phase into electrical signal? And how to assemble non-wafer optical components? Not mentioning the required post signal processing. The inability of doing that by a mass-production technology imposes a serious production bottleneck. And of course mass production reduces the total cost much over a one-by-one assembly. Also regarding the cost, many other approaches require expensive components too. For instance, a spectrum analyzer required for determining the spectral shift achieved by the interferometric structures is far more expensive than a quadrant photodiodes. So the cost analysis should be done on the whole solution, not on just one component in the system.

As for other non-optical devices, we already highlighted that the main reason of choosing an optical accelerometers at the first place over non-optical types is its immunity to ESD and EM. If the application doesn’t require that, the user can then choose whatever convenient.

As for the spring designs, the design compatible with the SOIMUM®s process –offered by MEMSCAP- produces rather good yield. We already fabricated design 1 by this process, and we got a yield of 22/28; which could has been better if we instructed MEMSCAP to not remove the dies mechanically from the dicing tape holding them, as the adhesive martial that hold the wafer during dicing may damage the structure. Instead, acetone, isopropanol, or any similar solution, can be used to dissolve the adhesive material first, then the dies can be removed and dried.

The point of alignment accuracy issue has been furtherly clarified in the revised manuscript in section 3.3.3. And the fabrication compatibility with the SOIMUMS process has been stated in section 2.1.

  1. Verification of the results is completely missing. The authors did not perform even a simplest proof of concept experiment to see if their concept is feasible and if the results are at least similar to the simulated ones, let alone a full experimental characterization of a finished and encapsulated device. They did not present a comparison with the literature data on similar structures (which do exist, as we have witnessed above).

The paper is only dedicated to presenting the concept, design and simulations of the device. We edited the title in the revised manuscript to reflect this fact to be “Design and Modeling of Fiber-Free Optical MEMS Accelerometer Enabling 3D Measurements". Experimental characterization of the mechanical part is now under development, but they can’t be finished within the 10 days period allowed by the journal to submit the revised manuscript. Early results show an error of only 16% in the mechanical sensitivity between the measurements and the simulations. As for a characterization of a finished and encapsulated device, this stage is rather industrial and hardly found in scientific publications; most of them do the characterization on a lab prototype like what we do.

As for the comparison, we presented a qualitative -rather than quantitative- comparison with the literature on similar structures of optical types regarding their principle of operation and the need of an optical fiber or non-standard assembly component, a complex DSP, or being only 1D, while our design overcomes these disadvantages.

There is a number of scientifically dubious claims and inconsistencies. I quote here only a few:

  1. Incorrect claims: the authors express their opinion that fiber-based solutions are inferior to their approach since they are too large (albeit their own solution is actually cm-sized) and since fibers will bend under acceleration (they do not mention that fibers can be immobilized – e.g. by gluing to a substrate and that, alternatively, waveguides can be monolithically integrated into the same chip as the rest of the device.

The fiber-based solutions are even larger than the size estimation stated in the manuscript, as presented above in Figure 1. One fiber connector is over the size of our device, even without connecting the source and detector. And if gluing is to be adopted instead of using connectors, that implies an assembly process not compatible to monolithic fabrication. The PI has worked for several years with alignment fibers, striping and gluing pare fibers to the substrate and has a good idea about their difficulty and very low yield. Not mentioning that the fibers and waveguides still needs to be coupled to the light sources and detectors. This is the real bottleneck as their coupling can’t be done by wafer bonders or any other cleanroom compatible microfabrication tools. In addition, the fiber core and waveguide widths are just few micrometers, which imply very precise alignment, usually from the side not vertically as done by wafer bonders. Any small misalignment for even fraction of micrometer affects the coupling severely. While the alignment in our case is far more tolerable as the areas to be aligned are in the millimeter ranges, while wafer bonders can offer accuracy in the micron and submicron range, as detailed above in comment 4.

  1. Incorrect claims: The authors write that there are few optical applications of accelerometers, while in reality there is a vast number of them, especially among those presented within last few years (the authors are advised to check the above mentioned reviews from 2020 and 2021)

Actually, we considered them few comparing to the electronic-based types. Nonetheless, the word “few” has been replaced by “numerous” in the revised manuscript.

  1. Incorrect claims: the authors write “it is expected that the size of the proposed device will be small compared to other optical accelerometers.” However, according to the authors themselves, the device dimensions are 1 cm × 1 cm × 2 cm. As far as the optical/photonic accelerometers are concerned, a majority of the photonic crystal-based devices are vastly smaller, while the standard, of-the-shelf non-optical devices usually take up a package of about 1 cubic millimeter.

As detailed above, the size 1 cm × 1 cm × 2 cm is an exaggerated estimation upon using already packaged components, while integrated die form is expected to be much lower. In addition, the other optical/photonic accelerometers are either 1D, or they are fiber-based solutions, or require sophisticated processing. As our device is 3D, its size should be compared by 3 of such single-axis accelerometers.

The size of the photonic crystal part only can be small indeed, but it can’t do the whole transduction up to the electrical signal by itself. Such photonic crystal-based devices are usually based on measuring the phase shift or the spectral response from the photonic crystal, which requires other parts to do so, not just a simple readout circuit. Thereby, the size of the phase transduction components or the spectrum analyzer required for these measurements should be counted as well. And they still need challenging alignment and coupling of light through optical fiber or waveguides. Not mentioning the size of the light source and photodiode, which can not be integrated and vertically bonded within the same package size as in our case. For that coupling, fiber connector might be needed, and the size of even one fiber connector is over the size of our device.

Indeed, non-optical devices may have less size, but they are out of our comparison.

We have furtherly clarified the issue of the size in the revised manuscript.

  1. Incorrect claims: the value of mechanical sensitivity (in mm/G or nm/G) is not the main figure of merit for a photonic accelerometer as the authors appear to think when they claim that the lower values of this parameter represent a disadvantage. Actually the overall sensitivity and device output will strongly depend on the readout method (for instance, interferometric readouts will effortlessly measure sub-nanometer displacements, so they will be able to utilize orders of magnitude smaller mechanical sensitivities and still obtain a high overall response).

Actually, we clearly stated that the total sensitivity is the multiplication of the mechanical sensitivity x optical sensitivity in equations 11 and 12. The stated values in the introduction were just to give the reader an idea about the previously achieved mechanical sensitivity in literature. And we never claimed that the low values of this parameter represent a disadvantage or made any comparison by them. On the contrary, we clearly stated that: “methods that used wavelength interference, reflected light position or change in the refractive index showed outstanding performance; nevertheless, they need digital signal processing (DSP), and they were complex, bulky and costly.” And no one can argue that measuring spectral responses is more complicated that just measuring a current value that directly reflect the acceleration value. To avoid such confusion, we removed all the numerical values obtained by previous literature, so that the reader don’t get the notion that we are holding any quantitative comparison.

  1. Incorrect claims: the authors write that their accelerometer “is monolithic in fabrication and assembly,” while actually it is a false information, since commercial components are used for the light source and for the detector, which definitely are not monolithically integrated with the rest of the accelerometer (seismic mass).

Commercial components are used for the light source and for the detector as we don’t have the facility to fabricate them on the wafer level, and we couldn’t order them from their manufacturers in dies form. This is because we are an academic institute, not an industrial one. However, such problem can be solved in the mass production scale as dials can be done with the appropriate manufacturers of such components to provide them as wafers. The structure itself doesn’t necessitate having of-the-shelf components or require pre-packaging of the part alone. To avoid any false notion, we changed such expressions in the revised manuscript to be “enables monolithic fabrication and assembly on the wafer level upon the availability of LEDs and detectors in the dies form”.  

  1. Inconsistencies: At the beginning of their article the authors argue that the competing piezoresistive accelerometers “suffer from thermal drift” and quote this as their main deficiency (lines 44, 45). However, for their own sensor they quote, “high-power LED may cause some heating in the device, this heating may cause a thermal drift in the photo-detector responsivity and the responsivity curve may change [25]. Hence, acceleration reading may have some errors.” (lines 403-405) – basically the same issue they quoted to eliminate vastly cheaper and simpler piezoresistive and capacitive devices.

We never claimed that our sensor overcomes the disadvantages of piezoresistive accelerometers, or any other non-optical accelerometers, except for less suitability of ESD and EM interference. The first paragraph of the introduction just presents a quick survey on the other different accelerometers, generally stating their advantages and disadvantages (as you kindly suggested yourself to keep their “pro et contra” in comment 14). Thereby there is no inconsistency, as there is nothing prevents different techniques from sharing same disadvantages, including ours. And we never specified anywhere in the article that our design is preferred for no thermal drift.

  1. Inconsistencies: Why use Comsol Multiphysics to simulate mechanical properties only, while the optical properties are determined by Matlab, when the both simulations can be done simultaneously in Comsol (after all, this is the main advantage of this FEM package – even its full name includes the word “Multiphysics”).

We could not combine the optical and mechanical parts of the device in one FEA simulation, as this needs a tremendous processing power and time. We actually tried the multiphysics solvers provided by COMSOL; however, the simulation took days and then crashes without returning any outputs, which we repeated many times with different conditions and got the same problem. This can be expected with the electromagnetic FEA simulations, as the number of meshing points required for simulation is massive. We need at least 8 points to represent one period of a sine wave. So, to simulate the propagation of light waves of 850 nm wavelength in a 0.5mmx3mmx3mm volume, this will require 4.2x10^16 points. Even using the less accurate ray optics will not make the processing power feasible. This is why we decided to separate the mechanical from optical modeling and used simplified analytical modeling for the optical part, as it is more complex from the processing point of view.

  1. Dubious claims: it appears that the authors implicitly assume that there are no scattering losses and no multiple reflections (e.g. from the walls, top and bottom of the encapsulating structure), while actually this could well be the main source of inaccuracies in their simulation. If they performed a full FEM simulation for the chosen geometry without any simplifications/approximations, they would have better data. I can only assume that such a simulation would necessarily be overly CPU-intensive.

Indeed, the reviewer is correct about the assumption that such simulations requires very demanding CPU capabilities. As stated in the previous point, we already tried such numerical simulations, and couldn’t obtain results. The meshing element size is in the hundreds of nanometer range, while the structure is in the millimeter range. Upon the testing and calibration of the device, such error can be estimated and subtracted from the readout values. We referred to such source of error in the revised manuscript and how it can be mitigated during the device calibration in the “3.3.3 Noise and resolution” subsection. 

  1. Article organization: the manuscript is overlong and needlessly repeats the well-known facts. The long text about the non-optical solutions could be almost completely skipped (especially in the vastly overlong Introduction, which, paradoxically, quite weakly handles the optical/photonic devices which are the main topic of the paper. Introduction reads like a very short review paper on conventional MEMS accelerometers. There is no need for the description of the operating principles of every single type of non-optical MEMS accelerometers since none of it contributes to further reading. Only the essentials and pro et contra arguments for their use should be kept. A glaring example of needless inclusion of excess material is met already in Abstract, where the authors elucidate in no less than 3 lines the basic principles of the operation of quadrant PIN photodiodes.

The introduction has been shortened in the revised manuscript according to this recommendation.

But we never elucidate the basic principles of the operation of quadrant PIN photodiodes neither in the Abstract, nor anywhere in the manuscript. Actually, the 3 lines in the Abstract explains the principle of operation of the entire proposed accelerometer, the quadrant photodiodes is only mentioned as a part of the device. And of course it is essential to describe the device in the Abstract.  

  1. English is rather readable throughout. This does not mean that it is without its share of errors in terminology and style. The authors should use “quadrants” instead of “panels” or “quarter detectors” for their quadrant photodiode. Also, they should not use the term “quadrature detector” (in a rather poorly written sentence “The used detector is a quadrature detector. Thus, it is treated as a four-isolated detector.”). They should rewrite “small slop”, “furtherly,” “ain,” “are be,” “a four-isolated detector,” “letral,” “mechnicle,” (repeated twice) “negligted,” “neglable,” “algorism,” etc.

We thank the reviewer for bring our attention to such typos. They all have been corrected in the revised manuscript.

  1. The source of eq. 6 is not cited. Any descriptions, derivations or clarifications of eq. 6 are missing.

We actually stated, the reference of Equations 6 at the beginning: "I. Moreno proposed an analytical method to model the behavior of square LED irradiance in the near field [21]."  Now it is reference [30] in the revised manuscript.  The descriptions and clarifications of Equations 6 can be found in this reference, as they are irrelevant to the topic.

  1. Nothing is mentioned about the existing solutions for 3D-sensitive spring systems in the state of the art part of Introduction, albeit this is a well thread and well-known area.

Indeed this is a well-known area, this is exactly why we didn’t elongate the introduction more by this part, especially that it is only a part of the device. And we added the reference we relied on in the corresponding mechanical design section (#2.1). Nonetheless, based on your recommendation, we referred to the design of the 3D-sensitive spring systems in the introduction, quoting another reference [26] in the revised manuscript.

  1. There is an error in Fig. 5: the two images are overlapped with what I expect was intended to be the inset.

Fig. 5 -now Fig. 6 in the revised manuscript- consists of an image for part (a) represents a 150 narrow-angle LED, and two images for part (b) represent a 600 wide-angle LED and the inset as a zoom on the short distance range.

  1. References: the article does include recent publications, however these are not the most relevant ones. Also, disproportionally many of them are dedicated to the conventional and well known MEMS accelerometers. For more recent references on photonic accelerometers, authors are advised to see the above quoted review papers, and the references cited therein.

We thank the reviewer for the suggested references. All the recommended papers mentioned above have been added to the References in the revised manuscript.

Reviewer 3 Report

This authors presented the design, simulation and modelling work of an accelerometer, capable of measuring acceleration in all three dimensions. The device replies on the spring movement pattern and corresponding change of power levels on the 4 PD segments to interrogate the acceleration direction and amplitude. I believe the idea is novel and the modelling is sound. This work is suitable for publication in Micromachines, after the following issues are cleared.

1)  The authors claim this optical accelerometer is immune to ESD as it uses light signal to detect the acceleration. However, the PD converts light power into current and still requires the processing of electronic signals. The circuits may still be affected by ESD. The authors need to clarify this in detail.

2) As the springs will cause oscillation, resulting in periodic change of photo current, how will this affect the accelerometer performance?

Author Response

Response to the Reviewer’s Comments and Suggestions (Reviewer 3)

Dear Editors and Reviewers,

The authors would like to thank the reviewers for their knowledgeable criticism and wise recommendations towards the improvement of our manuscript. We are going to give answer to their comments and recommendation point-by-point and include the edits in the revised manuscript (Answers are typed in red).

My Best Regards,

Samir AboZyd

Comments and Suggestions for Authors (Reviewer 3)

  1. The authors claim this optical accelerometer is immune to ESD as it uses light signal to detect the acceleration. However, the PD converts light power into current and still requires the processing of electronic signals. The circuits may still be affected by ESD. The authors need to clarify this in detail.

Yes indeed, the electronics interfacing circuit will still have ESD susceptibility.  However, the mechanical-optical transduction of the sensor itself will not be affected. On the other hand, accelerometers that depend on the capacitance change or electrical resistance change are sensitive to ESD in both the sensing part and electrical interfacing part. Therefore, per your suggestion, we changed the expression of “immune to ESD" to be "less susceptible to ESD" in the revised manuscript.

  1. As the springs will cause oscillation, resulting in periodic change of photo current, how will this affect the accelerometer performance?

Indeed, some unfavorable oscillations may happens in the mass-spring system if it is operated close to the mechanical resonance frequency, as shown in the below figure, or when working in the under damping mode [1]. Therefore, to avoid such oscillations, we decided to work within a Mechanical Bandwidth of a maximum frequency of 244 Hz, which is less than two-thirds of the mechanical resonance frequency as specified in the manuscript. Moreover, upon adjusting the accelerometer to work in the critical damping or slightly over damping mode, the springs will only move with the same frequency of the input acceleration to be measured (within the range from 0Hz to 244 Hz) without any other unfavorable oscillations. And such low frequencies can be traced easily by the photodetector and the electronic circuit.

[1]          Kaajakari, V. Practical MEMS; Small Gear Pub: Las Vegas, Nev., 2009; ISBN 978-0-9822991-0-4.

Round 2

Reviewer 1 Report

In new revision of the manuscript, most of issues have been resolved. However, I still have some concerns about whether this work can be published.

In my comment #2, I asked authors to unify the characterization of the accelerometers, mentioned in Introduction, for easier comparison of the devices based on different operational principles. In response, authors wrote that “this is not an easy task for accelerometers sensitivity units, as every accelerometer has different physical characteristics and different output signal that cannot be represented by or converted to unified units” and in new version of the paper, they simply “removed all the numerical values obtained by previous literature, so that the reader doesn’t get the notion that we are holding a quantitative comparison”.
      I don’t consider this as proper solution, as reader now has no any idea about the advantages and disadvantages of different accelerometers, and can not even estimate if the device developed by authors is better or worse than others. Indeed, the output signals from accelerometers of different kinds may have different nature and different measurement units. However, the unitless dynamic range, or ratio between the signal range and its resolution or noise level, can be estimated and easily compared.

In comments 3 and 4, I requested the information about materials and fabrication processes that are supposed to be used in the implementation of this accelerometer. Partially this request was fulfilled. Nevertheless, it is still unclear why “the SOIMUMS processes by MEMSCAP are suitable for such fabrication”? Could authors give a little bit more details providing a general explanation of the advantages of their choice? A short description of these processes might be useful as it would allow to better understand why they satisfy the production requirements. Additionally, I think that it is necessary to show the dimensions for the proof mass and its internal chamber in Figure 2 to avoid any doubts about the accelerometer size. From further reading it becomes clear that these dimensions are about 0.3-1 mm, however, it is better to show the numbers that authors use in their simulations.

Finally, I still consider that this project should be completed by demonstration of the experimental prototype. As I mentioned previously in my comment #11, the estimates of the characteristics obtained purely form the design and computer simulation can not be compared with specifications of really existing devices. This is simply because it is virtually impossible to predict and simulate every problem that may emerge during real development and implementation process, and how this problem would affect final device parameters.
        From my vision, current work presents only a preliminary research phase, and its results can be published only if respected authors, with this paper, can convince readers of the potential benefits and enormous advantages of their proposed design.

Author Response

Response to the Reviewer’s Comments and Suggestions (Reviewer 1)

Dear Editors and Reviewers,

The authors would like to thank the reviewers for their knowledgeable criticism and wise recommendations towards the improvement of our manuscript. We are going to give answer to their comments and recommendation point-by-point and include the edits in the revised manuscript (Answers are typed in red).

My Best Regards,

Samir AboZyd

Comments and Suggestions for Authors (Reviewer 1)

  1. In my comment #2, I asked authors to unify the characterization of the accelerometers, mentioned in Introduction, for easier comparison of the devices based on different operational principles. In response, authors wrote that “this is not an easy task for accelerometers sensitivity units, as every accelerometer has different physical characteristics and different output signal that cannot be represented by or converted to unified units” and in new version of the paper, they simply “removed all the numerical values obtained by previous literature, so that the reader doesn’t get the notion that we are holding a quantitative comparison”.
          I don’t consider this as proper solution, as reader now has no any idea about the advantages and disadvantages of different accelerometers, and can not even estimate if the device developed by authors is better or worse than others. Indeed, the output signals from accelerometers of different kinds may have different nature and different measurement units. However, the unitless dynamic range, or ratio between the signal range and its resolution or noise level, can be estimated and easily compared

The resolution value of each device has been added in the revised manuscript with the unit of G to enable the comparison between the devices. For the references that didn’t mention such values, we either calculated it for them depending on their design parameters, or estimated it with a reasonable estimation based on their measurement technique.

  1. In comments 3 and 4, I requested the information about materials and fabrication processes that are supposed to be used in the implementation of this accelerometer. Partially this request was fulfilled. Nevertheless, it is still unclear why “the SOIMUMS processes by MEMSCAP are suitable for such fabrication”? Could authors give a little bit more details providing a general explanation of the advantages of their choice? A short description of these processes might be useful as it would allow to better understand why they satisfy the production requirements.

Actually, we had to send our design to MEMSCAP for fabrication as our cleanroom is temporarily not available due to moving our campus. But we think that such challenge can be faced by fabless companies and many research institutes because a custom fabrication facility can not be always available. So we believe that it is an advantage for our design to be compatible with such public fabrication service with reasonable cost as it a “Multi User MEMS Process”, so the cost is shared by several users. 

The clarification for the choice of SOIMUMPs process, in addition to a more detailed fabrication process, has been added to the revised manuscript.

  1. Additionally, I think that it is necessary to show the dimensions for the proof mass and its internal chamber in Figure 2 to avoid any doubts about the accelerometer size. From further reading it becomes clear that these dimensions are about 0.3-1 mm, however, it is better to show the numbers that authors use in their simulations.

 The dimensions of the proof mass-spring system have been added in Figure 2 in the revised manuscript. And the dimensions of the proof mass and its internal chamber have been added as text in the simulation section 3.1.

  1. I still consider that this project should be completed by demonstration of the experimental prototype. As I mentioned previously in my comment #11, the estimates of the characteristics obtained purely form the design and computer simulation can not be compared with specifications of really existing devices. This is simply because it is virtually impossible to predict and simulate every problem that may emerge during real development and implementation process, and how this problem would affect final device parameters.

We agree that making a prototype and getting experimental results are highly more reliable than simulation and analytical modeling. Nevertheless, the importance of simulation and analytical modeling in the research process cannot be denied. Getting good results in simulation and modeling is a necessary step before starting the more costly prototyping process. As, the prototype of the proposed accelerometer requires the fabrication of the mechanical MEMS part, purchasing the QPD and LED, and then bonding the three of them together. Moreover, it requires the designing of an ASIC interfacing to transduce the signal of the QPD into acceleration output. In fact, we did the first step of the prototyping and fabricated the MEMS mechanical part using the SOIMUMP®s process. We also tested its mechanical lateral sensitivity of design 2 (the design with the maximum sensitivity) using a simple testing setup and found it is close to the simulation with an error of less than 18%.However, this test is not mature enough for publication as it is only for lateral mechanical sensitivity, and only for design 2; so it doesn’t reflect the total device performance. We believe that such rather high error is due to the poor capabilities of the available imaging equipment, and therefore we are trying to secure more fund for upgrading our equipment to get better experimental results, then completing the next steps of the prototyping. But this of course takes more time  than the 8 days specified by the journal to submit our revised manuscript. We also believe that publishing the current work will vastly help us in securing the needed fund, and may even enable us establishing collaboration with other partners to extend our capabilities. If not, then the concept and foundation will be presented for other researchers to build upon as well.

  1. From my vision, current work presents only a preliminary research phase, and its results can be published only if respected authors, with this paper, can convince readers of the potential benefits and enormous advantages of their proposed design.

Based on the recommendation of the respected reviewer, we tried to convince the readers more in the revised manuscript with the potential benefits and advantages of the proposed design. We furtherly clarified that the optical accelerometers, generally, have some advantages that give them a position in the market. In addition to their inherited immunity to the magnetic field and ESD, they usually have higher sensitivity and resolution than most other kinds of accelerometers. According to our calculation, which of course needs experimental results to be final, our accelerometer can achieve 56-µG resolution, while, for instance, the typical capacitive accelerometers acceleration range starts from 1 mG [1]. Our design adds additional advantages over the current commercial optical accelerometers regarding the size and the compatibility with mass production and assembly because it eliminate the need of optical fibers. For instance, 

The image below presents one of the optical accelerometers in the market; and as can be noticed, the optical fibers and their 3D configuration are not as a single microchip.  The proposed optical accelerometer is the first optical accelerometer, up to our knowledge, that enables achieving 3-axial sensing on a single compact micromachined chip. Moreover, the fact it does not include optical fibers gives its fabrication and assembly an advantage that cannot be attained by any other fiber-based optical accelerometer. To illustrate, each part can be fabricated on wafers by normal cleanroom-based processes, then they can be assembled with high accuracy using a wafer bonder, without the need for the troublesome alignment process that other fiber-based optical accelerometers suffer from. Not mentioning the need to couple the optical sources and detectors to the fiber, while ours has the source and detector already integrated. Noting that the optical fiber core or even the optical waveguide widths are just a few micrometers, which implies very precise alignment. Any small misalignment for an even a fraction of micrometer affects the coupling severely. Not to mention that the coupling is usually from the sides not vertically, which can’t be done by wafer bonders or any other cleanroom compatible microfabrication tool. While the alignment in our case is far more tolerable. Many bonders can offer high alignment accuracy, such as BA8 Gen4 that offers up to 0.25 μm for top-side alignment and 1 μm for bottom-side alignment [SUSS MicroTec BA8 Gen4 Pro Bond Aligner Available online: https://www.suss.com/en/products-solutions/wafer-bonder/ba8-gen4-pro (accessed on 20 January 2022).]. Even less sophisticated systems such as EVG®610 BA is ± 2 µm for backside alignment, and ± 1 µm for transparent alignment [EVG®610 BA Semi-automated Bond Alignment System Available online: https://www.evgroup.com/products/bonding/bond-alignment-systems/evg610-ba/(accessed on 20 January 2022)]. This is very convenient for our case as the area of the LED is 1 mm x1 mm, the proof mass is also 1 mm x1 mm, and each quadrant of the detector is 1.44 mm x1.44 mm. Even the small errors arise from such misalignment can be mitigated during the device calibration.

Moreover, the size of the proposed accelerometer is expected to be smaller than most other optical accelerometers, and to be smaller than all the 3-axial configurations that need the assembly of three single-axial optical accelerometers.  Furthermore, some of these alternative optical approaches, even those in the research phase, also require sophisticated post-processing of the measured data to obtain the acceleration values and even require a non-integrable lab setup, such as those based on measuring the spectrum. So such approaches can not be developed into a practical solution in the form of a stand-alone module that can be directly integrated within the needed application requiring just electrical connections; while our approach can offer that.

Luna’s os7500 accelerometer series [https://lunainc.com/sites/default/files/assets/files/resource-library/Distributed%20Vibration%20Monitoring%20using%20FO%20Accelerometers.pdf. Accessed: 22 Jan. 2022].

[1] Microelectronics and Signal Processing.; Goel, S., Ed.; CRC Press., 2021; ISBN 978-1-00-316822-5.

Reviewer 2 Report

The revised version of the manuscript is vastly improved. The authors responded apropriatelly to most of my comments and introduced the necessary modifications and elucidations. It is also a good thing that they decided to modify the title to better reflect the manuscript contents.

I still think that the basic premise is very slight on originality, being actually a combination of two well-known concepts (LED-quad diode optical pair and a spring system with 3 degrees of freedom) and that the contribution is incremental, but at least the concept has a virtue of simplicity and is very meticulously and expertly analyzed and elaborated in depth, which makes it publishable.

I believe that in the present form the manuscript is acceptable for Micromachines journal after minor revisions.

I wish the authors success in the rather formiddable future task of implementing their concept experimentally!

The following list of relatively minor issues that are readily solvable is meant to facilitate the remaining work to the authors:

1. The authors omitted to mention an additional benefit of their approach, and thus unintentionally harmed their own text: their spring system is applicable to many (not all) other accelerometers with optical readout as well! A sentence stating this should be included in Conclusion.

2. I advise the authors to insert the following additional introductory sentence at the very beginning of subsection 2.1: “In order to explain the design and modelling of the mechanical part of the accelerometer, a few introductory words are needed about our planned MEMS fabrication process.” The way this part is written now, the paragraph about fabrication is out of place because the subsection is dedicated to design and modelling.

3. Scientific terminology: the authors use the term “monolithic” inproperly. Its meaning is “manufactured within a single crystal”. A MEMS system produced by bonding three chips is not monolithic, but hybrid. I suggest the correction of this claim throughout.

4. Although English is vastly improved, the manuscript nevertheless retains some flaws, for instance sentences that begin with the word Because and Which. Also, the text “3D-sensitive spring systems in a” should be “3D-sensitive spring systems is a” (line 148), “three different springs design” is “three different springs designs” (line 190), “trying to compromise” should be “trying to find a trade-off” (lines 211, 212), “maintaining the mechanical sensitivity to an acceptable value” should be “maintaining the mechanical sensitivity at an acceptable value” (lines 234, 235), “acceptable amount of intensity” should be “acceptable intensity”, “which compromise” should be “which represents a trade-off” (line 357); the authors should delete the repeating part of the sentence “The spring constant is proportional to the mechanical resonance frequency” (lines 362, 363), “implemented on” should be “implemented in”, “rang” should be “range” (line 405), “Kelven” should be “Kelvin” (421), “temerature” should be “temperature” (423), “Therfore” is “Therefore” (440), “negligable” is “negligible” (445), “and less susceptible” is “and is less susceptible” (495).

5. The description of the operating principle is written twice in 2 different places, in the 1st paragraph of Section 2 (lines 161-164) and in the 1st paragraph of 2.2 (242-246).

6. Please add the reference for eq. 6 in the line 273.

Author Response

Response to the Reviewer’s Comments and Suggestions (Reviewer 2)

Dear Editors and Reviewers,

The authors would like to thank the reviewers for their knowledgeable criticism and wise recommendations towards the improvement of our manuscript. We are going to give answer to their comments and recommendation point-by-point and include the edits in the revised manuscript (Answers are typed in red).

My Best Regards,

Samir AboZyd

Comments and Suggestions for Authors (Reviewer 2)

Before starting to reply to the precious comments and suggestions, I want to express my thanks for your supportive and kind words.

  1. The authors omitted to mention an additional benefit of their approach, and thus unintentionally harmed their own text: their spring system is applicable to many (not all) other accelerometers with optical readout as well! A sentence stating this should be included in Conclusion.

Thank you for your useful suggestion. The recommended statement has been added in the Conclusion in the revised manuscript.

  1. I advise the authors to insert the following additional introductory sentence at the very beginning of subsection 2.1: “In order to explain the design and modelling of the mechanical part of the accelerometer, a few introductory words are needed about our planned MEMS fabrication process.” The way this part is written now, the paragraph about fabrication is out of place because the subsection is dedicated to design and modelling.

 Thank you for driving our attention to this point. The recommended sentence has been added in the revised manuscript.

  1. Scientific terminology: the authors use the term “monolithic” inproperly. Its meaning is “manufactured within a single crystal”. A MEMS system produced by bonding three chips is not monolithic, but hybrid. I suggest the correction of this claim throughout.

The word "monolithic' has been edited to hybrid per your suggestion.

  1. Although English is vastly improved, the manuscript nevertheless retains some flaws, for instance sentences that begin with the word Because and Which. Also, the text “3D-sensitive spring systems in a” should be “3D-sensitive spring systems is a” (line 148), “three different springs design” is “three different springs designs” (line 190), “trying to compromise” should be “trying to find a trade-off” (lines 211, 212), “maintaining the mechanical sensitivity to an acceptable value” should be “maintaining the mechanical sensitivity at an acceptable value” (lines 234, 235), “acceptable amount of intensity” should be “acceptable intensity”, “which compromise” should be “which represents a trade-off” (line 357); the authors should delete the repeating part of the sentence “The spring constant is proportional to the mechanical resonance frequency” (lines 362, 363), “implemented on” should be “implemented in”, “rang” should be “range” (line 405), “Kelven” should be “Kelvin” (421), “temerature” should be “temperature” (423), “Therfore” is “Therefore” (440), “negligable” is “negligible” (445), “and less susceptible” is “and is less susceptible” (495).

“Because” has been changed to “To illustrate” or omitted from the beginning of the sentence.

“.Which” is changed to “, which”

“3D-sensitive spring systems in a” has been changed to“3D-sensitive spring systems is a”

“three different springs design” has been  changed to “three different springs designs”

“trying to compromise” has been changed to “trying to find a trade-off”

“acceptable amount of intensity” has been changed to “acceptable intensity”

“which compromise” has been changed to “which represents a trade-off”

the repeated part is the sentence “The spring constant is proportional to the mechanical resonance frequency”  has been deleted

“implemented on” has been changed to “implemented in”

“rang” has been changed to “range”

“Kelven” has been changed to “Kelvin”

“temerature” has been changed to “temperature”

“Therfore” ” has been changed to “Therefore”

“negligable” has been changed to “negligible”

“and less susceptible” has been changed to “and is less susceptible”

  1. The description of the operating principle is written twice in 2 different places, in the 1stparagraph of Section 2 (lines 161-164) and in the 1stparagraph of 2.2 (242-246).

 The description in the 1st paragraph of 2.2 is omitted in the revised manuscript.

  1. Please add the reference for eq. 6 in the line 273.

The reference for eq. 6 has been moved just before the equation as per your kind recommendation.

Round 3

Reviewer 1 Report

Current version of the paper appears to be much advanced in comparison with previous ones, and, I believe it can be published with some minor changes.

Considering scientific and engineering aspects of this work, I think that one of important characteristics of the suggested accelerometer still can be improved. I mean the consumed electrical power. Indeed, taking that the resolution of this sensor is limited by mechanical noise, because the contribution of photoelectric shot-noise is much lower, why isn’t it possible to significantly reduce the optical power by at least one order of magnitude? The shot-noise-limited resolution will increase, of course, but it still will be less than the resolution defined by the mechanical noise. Is it correct?

Text copy editing is also required:

Page 5, line 212: “cusmized fabrication” should read as “customized fabrication”.

The equation (3) on page 7 needs better math typesetting.

Author Response

Response to the Reviewer’s Comments and Suggestions (Reviewer 1)

Dear Editors and Reviewers,

The authors would like to thank the reviewers for their knowledgeable criticism and wise recommendations towards the improvement of our manuscript. We are going to give answer to their comments and recommendation point-by-point and include the edits in the revised manuscript (Answers are typed in red).

My Best Regards,

Samir AboZyd

Comments and Suggestions for Authors (Reviewer 1)

  1. Considering scientific and engineering aspects of this work, I think that one of important characteristics of the suggested accelerometer still can be improved. I mean the consumed electrical power. Indeed, taking that the resolution of this sensor is limited by mechanical noise, because the contribution of photoelectric shot-noise is much lower, why isn’t it possible to significantly reduce the optical power by at least one order of magnitude? The shot-noise-limited resolution will increase, of course, but it still will be less than the resolution defined by the mechanical noise. Is it correct?

We agree that decreasing the optical power by one order of magnitude will only increase the optical noise from about 5 μG to 16 μG and the total noise from 56.2 μG to 58.2 μG. However, it will also decrease the sensitivity from 156 μA/G to about 15.6 μA/G. This may render the design of the electronic interfacing circuit a bit more complex, especially if the accelerometer is mainly intended for measuring low acceleration amplitudes.  Nevertheless, we think that your suggestion can provide a good compromise for the applications that may require lower power consumption. Therefore, we have mentioned this compromise in the revised version of the manuscript in section 3.3.3.

  1. Page 5, line 212: “cusmized fabrication” should read as “customized fabrication”.

The word “cusmized” has been edited  to “customized” in the revised manuscript.

  1. The equation (3) on page 7 needs better math typesetting.

The equation (3) has been formatted in the revised manuscript.
